# Auditing GeoFM Evaluation for Field-Extent Segmentation: Label Proxies, Baselines, and When Frozen Features Match Fine-tuning

## Abstract

Published rankings of geospatial foundation models (GeoFMs) on agricultural field-extent segmentation rest on two common evaluation choices. First, when polygon-derived field labels are scarce, an ESA WorldCover "cropland" land-cover product is used as a stand-in; on identical Sentinel-2 pixels across six countries, this proxy lifts a per-pixel random forest from 0.55–0.82 to 0.79–0.96 AUROC, making it foundation-model-competitive while the true field-extent task is unchanged. Second, published comparisons often use per-pixel spectral baselines; a from-scratch U-Net on the same true labels reaches 0.89–0.98 AUROC in all six regions, has higher point AUROC than a Prithvi frozen probe in five of six regions (India remains the frozen-probe AUROC exception), and wins IoU outside Cambodia in the ten-seed decoder comparison. PANGAEA reports a similar supervised-baseline pattern on AI4SmallFarms. Our claims are limited to single-date, per-pixel field-extent segmentation rather than parcel-boundary or instance delineation.

With both evaluation choices addressed, we ask whether full end-to-end fine-tuning of the GeoFM backbone is necessary. In a single-recipe comparison across six countries, two GeoFMs, and ten matched training seeds, a frozen backbone with a trained decoder is TOST-equivalent to full fine-tuning at $\varepsilon = 0.02$ AUROC in 9 of 12 region×model cells (Prithvi equivalent outside Kenya; TerraMind 4 of 6), with a stable failing set across $\varepsilon \in [0.02, 0.03]$ (Appendix Table 9). Kenya is the consistent sparse-positive exception: full fine-tuning wins for both GeoFMs with CIs excluding zero. TerraMind India is non-equivalent in the frozen-favored direction because the CI extends past the equivalence margin. Cross-region transfer is model- and regime-dependent: on the full 30 directed transfer matrix a from-scratch U-Net is strongest (0.826 mean AUROC), both frozen GeoFMs trail it, and Prithvi frozen trails Prithvi full fine-tuning by 0.013 AUROC. In an *exploratory* post-hoc partition excluding Kenya (20 transfers), both frozen GeoFMs beat their full-FT counterparts, and TerraMind frozen is statistically indistinguishable from the U-Net. The frozen decoder trains about 110× fewer parameters than full fine-tuning. We report the sparse-positive Kenya boundary case and provide a ten-point evaluation protocol.

## 1 Introduction

GeoFM field-mapping benchmarks can change their apparent conclusion through evaluation choices that are easy to miss: whether the target is polygon-derived field extent or a cropland proxy, and whether the non-FM baseline is a weak per-pixel spectral model or a task-specific supervised segmentation model. We audit both choices on identical Sentinel-2 pixels across six countries, and then use the corrected setup to ask whether the standard practice of full end-to-end fine-tuning is necessary. For single-date field-extent segmentation, once both evaluation choices are addressed and the frozen comparator is capacity-matched, frozen features are competitive.

Two conveniences shape many benchmarks in this area. The first is a land-cover product (ESA WorldCover "cropland" at 10 m) used as ground truth because field-polygon annotations are scarce. The second is the

strength of the non-FM baseline: per-pixel spectral models (random forests on band values) are common, but a matched-pixel from-scratch U-Net is not always reported alongside. Both choices change model ranking; this paper quantifies the impact and asks what survives once both are addressed.

Section 4 identifies two evaluation choices that can bias these benchmarks. The first is the label choice: swapping polygon-derived field-extent labels for the WorldCover proxy raises a per-pixel random forest from near chance to GeoFM-competitive on identical pixels and reverses the apparent size bias. The second is baseline strength: a supervised U-Net trained from scratch on true labels reaches 0.89–0.98 AUROC in every region and is competitive with frozen GeoFMs outside India; on IoU, the U-Net wins outside Cambodia in the ten-seed decoder comparison. External evidence supports the two concerns separately: PANGAEA (Marsocci et al., 2024) reports a similar supervised-baseline pattern on AI4SmallFarms, while the Prithvi-EO-2.0 report's published comparison is a multi-temporal crop-segmentation task rather than a polygon-derived field-extent benchmark (Szwarcman et al., 2024).

Section 5 compares frozen-backbone GeoFM features (decoder trained on frozen features) against full end-to-end fine-tuning in two settings. In-region, across six regions, two GeoFMs (Prithvi and TerraMind), and ten matched seeds per cell, the frozen backbone is TOST-equivalent to full fine-tuning in 9 of 12 region×model cells at $\varepsilon$=0.02 AUROC. Cross-region (train on one country, test on another with no target-region labels), the full 30 directed-transfer matrix gives a more cautious result: a from-scratch U-Net is strongest overall, while frozen GeoFM features beat their full-FT counterparts on an exploratory non-Kenya subset and TerraMind frozen is statistically tied with the U-Net there. Combined, the results support testing a frozen backbone with a decoder before paying the cost of full fine-tuning, while retaining a strong supervised baseline and checking explicitly for regimes like the Kenya boundary case. Section 6 reports the Kenya failure case (around 0.1% field-pixel coverage). Section 7 quantifies the parameter savings. Section 8 gives a ten-point protocol.

**Contributions.** (1) A same-pixel evaluation setup that separates the effect of label choice from that of baseline strength on GeoFM evaluation, applied across six countries with Fields-of-The-World boundaries. (2) Evidence that those two choices shift the reported GeoFM advantage on field-extent tasks, with PANGAEA corroborating the supervised-baseline finding. (3) The corrected-evaluation answer to the fine-tuning question: a single-recipe $n$=10 finding that full fine-tuning is often equivalent to a capacity-matched frozen-decoder in-region, that frozen decoders beat full fine-tuning on an exploratory non-Kenya transfer subset, and that this equivalence holds with a roughly 110× trainable-parameter gap, with extreme label sparsity (Kenya) as a documented boundary condition. (4) A ten-point evaluation protocol and a released, audited artifact.

## 2 Related Work

**Geospatial foundation models.** Recent GeoFMs pretrain on large unlabeled satellite archives and are reused across tasks: masked-autoencoder models (SatMAE (Cong et al., 2022), Scale-MAE (Reed et al., 2023), Prithvi-EO (Szwarcman et al., 2024)), contrastive and multimodal models (CROMA (Fuller et al., 2023), Clay (Clay Foundation, 2024), TerraMind (Jakubik et al., 2025), AnySat (Astruc et al., 2024)), wavelength-conditioned models (DOFA (Xiong et al., 2024)), spectral models (SpectralGPT (Hong et al., 2024)), pixel-timeseries models (Presto (Tseng et al., 2023)), and large supervised pretraining (SatlasPretrain (Bastani et al., 2023)). Their downstream evaluations almost always fine-tune the backbone. We test that default for field-extent segmentation.

**Frozen features vs. fine-tuning.** Outside Earth observation, a substantial literature documents that frozen pretrained features with a light head are often competitive with full fine-tuning, and that fine-tuning can harm out-of-distribution performance. Kumar et al. (2022) show that fine-tuning distorts pretrained features and underperforms frozen linear probing under distribution shift, which is the pattern we observe across regions. Locked-feature training (Zhai et al., 2022), transfer studies (Kornblith et al., 2019; Kolesnikov et al., 2020), and the linear-probe evaluation of self-supervised models (He et al., 2022) all point in the same direction. We find evidence consistent with the same pattern for geospatial field-extent segmentation, with an explicit boundary condition and a cost analysis.

**What is not domain confirmation.** The direction of our frozen-vs-fine-tuning result echoes Kumar et al. (2022) and the linear-probe literature, and we credit that lineage. What that literature does not determine is (a) whether the same recommendation holds in a GeoFM setting where downstream evaluations almost uniformly fine-tune and the standard frozen comparator is a linear probe rather than a capacity-matched decoder, (b) whether it survives extreme positive-label sparsity, and (c) whether it survives the label choices GeoFM benchmarks actually make. First and centrally, we document a sparse-positive failure regime with no counterpart in the ID/OOD framing of Kumar et al. (2022): at Kenya's ∼0.1% positive-pixel rate and 155 training chips, the frozen decoder collapses (AUROC drop of 0.10–0.12 vs. full fine-tuning for both GeoFMs; threshold-metric collapse to near-all-negative predictions) and adaptation is necessary (Section 6). This is a label-geometry failure mode specific to sparse-annotation geospatial tasks and it bounds the frozen-feature recommendation. Second, the GeoFM literature's standard published comparison confounds backbone adaptation with decoder capacity: fine-tune-plus-decoder against a frozen *linear* probe. Giving the frozen backbone the same 2.75M-parameter decoder as the fine-tuned model removes the gap in most in-region cells (Section 5); the "fine-tuning helps" pattern in published GeoFM benchmarks may in part be a decoder-capacity artifact. Third, the comparison itself is only meaningful once the label audit is done: under the WorldCover proxy, the task becomes spectrally trivial and all configurations saturate above 0.90 AUROC (Table 2), so importing the frozen≈fine-tuned result into geospatial benchmarks without a label control measures the wrong task. Our contribution is therefore not that frozen features can match fine-tuning, which is already established, but the geospatial-specific characterization of when they do and where they fail.

**EO benchmarks and evaluation.** GEO-Bench (Lacoste et al., 2023) and PANGAEA (Marsocci et al., 2024) standardize GeoFM evaluation; we use PANGAEA's results as external corroboration. Fields of The World (Kerner et al., 2024) supplies the field polygons that let us replace land-cover proxies. Spatial autocorrelation is a well-known evaluation pitfall in geospatial ML: train and test chips drawn from nearby tiles can inflate scores. We use a chip-grouped split (no chip in both partitions) and report tile-disjoint sensitivity (Appendix Table 17) to address it. We audit the evaluation choices (labels and baselines) rather than propose a new model.

## 3 Setup

**Data.** We use Sentinel-2 L2A chips paired with field polygons from Fields of The World (Kerner et al., 2024) across six countries spanning smallholder to industrial agriculture: India, Cambodia, Vietnam, Kenya, France, and the Netherlands. For each chip we form two labels on the same pixels. The TRUE label is 1 when the pixel lies inside an FTW field polygon, rasterized to the chip grid in the chip's own CRS. The PROXY label is 1 when the pixel is ESA WorldCover "cropland" (class 40, (Zanaga et al., 2022)). The task throughout is per-pixel binary field-extent segmentation (inside-vs-outside a field polygon); we do not evaluate boundary delineation or instance separation between adjacent parcels, and we use "field mapping" as a synonym for this per-pixel task.

**Models.** Non-FM baselines: a per-pixel random forest on the 12 raw S2 bands, a $5 \times 5$ windowed random forest (300-d), and a from-scratch U-Net (Ronneberger et al., 2015) (31M parameters). GeoFMs: Prithvi-EO-2.0-300M (Szwarcman et al., 2024) and TerraMind (Jakubik et al., 2025) (with Clay and AnySat for breadth), used in two modes. Frozen: backbone weights fixed, then either a linear probe or a trained decoder is fit on frozen features. Fine-tuned: backbone trained end-to-end with the same decoder.

**Nomenclature.** We use three terms consistently. **Frozen-probe** = backbone weights fixed, a linear classifier ($\sim 10^3$ parameters) trained on extracted per-pixel features; Table 2 reports Prithvi consistently, with the four-model breakdown in Appendix Table 7. **Frozen-decoder** = backbone weights fixed, a 4-stage convolutional decoder (2.75M parameters) trained on extracted feature maps; this is the apples-to-apples comparator against full fine-tuning and is the configuration used throughout Sections 5–7 unless noted. **Full-FT** = backbone and the same decoder trained jointly end-to-end (307M trainable parameters). The three configurations can disagree: in Kenya, the best frozen probe across four FMs reports 0.853 AUROC

(AnySat; separate frozen-probe sensitivity), while the ten-seed frozen-decoder on Prithvi reports 0.653; we report both with the configuration explicit and discuss the gap in Section 6.

Table 1: **Model configurations used in this paper.** Five configurations appear across the results tables; each row names the trainable component, trainable-parameter count, whether the backbone is updated, the training budget, and which tables report each configuration.

| Configuration | Trainable params | Backbone updated | Optimizer / epochs | Reported in |
|---|---|---|---|---|
| per-pixel / windowed RF | (fitted trees) | — | sklearn defaults | Tables 2, 5 |
| frozen-probe (GeoFM + linear) | $\sim 10^3$ | no | logistic regression | Tables 2, 5, 7 |
| frozen-decoder (GeoFM + 4-stage conv) | 2.75M | no | AdamW, 80 epochs | Tables 3, 4, 8 |
| full-FT (GeoFM + same decoder) | 307M (Prithvi) | yes | AdamW, 80 epochs | Tables 3, 4, 8 |
| U-Net (from scratch) | 31M | n/a | Adam, 150 epochs | Tables 2, 3, 4 |

**Protocol and statistics.** All models are scored on the same held-out test pixels via a chip-grouped split (no chip in both train and test; overlap $= 0$; seed 20260514, 25% test). The main frozen-decoder, full-FT, and U-Net comparisons use a single-recipe ten-seed grid with ten matched training seeds per region and one software recipe; we report mean and a seed-paired $t$-interval at $df{=}9$ for matched frozen-decoder vs. full-FT differences (the same paired analysis used in Table 3 and Appendix Table 8). Frozen-probe AUROC and F1 in Table 2 carry chip-level 95% bootstrap CIs ($B{=}1000$) recomputed over the test chips, available in five regions (Cambodia per-chip CIs not recomputed; point estimates only). Equivalence claims use two one-sided tests (TOST; Lakens, 2017) with declared margin $\varepsilon{=}0.02$ AUROC in-region; Holm correction across the 12 primary TOST p-values retains all 9 equivalent cells at $\alpha{=}0.05$ (`ftw_inregion_equivalence.json`).

**TOST margin justification.** The margin is chosen for three reasons. First, the equivalence count is invariant across the operational range $\varepsilon \in [0.020, 0.030]$: the failing set (Prithvi–Kenya, TerraMind–India, TerraMind–Kenya) does not change over this range, so no conclusion depends on the exact margin (Appendix Table 9). Second, 0.02 AUROC is operationally negligible for field-extent mapping under class-imbalanced Sentinel-2 imagery: it is well below any difference we treat as substantive elsewhere in the paper (the smallest between-model gap we discuss is roughly 0.04 AUROC, the India frozen-probe-over-U-Net lead). Third, the direction of sensitivity to tighter margins is benign in the operational range: at $\varepsilon \in [0.010, 0.020)$ the only cells that leave the equivalent set are Prithvi–Netherlands and TerraMind–Netherlands, both in the *frozen-favored* direction ($\Delta = +0.013$ and $+0.009$ AUROC respectively), i.e., stricter margins in this range only reclassify cells where the frozen decoder already beats fine-tuning. Only at the strictest $\varepsilon = 0.005$ do two additional FT-favored cells (Prithvi–India, TerraMind–Cambodia) become formally non-equivalent, but their delta magnitudes (0.004 and 0.006 AUROC respectively) are below the operational tolerance that motivates the margin choice. Anchoring against pipeline noise: outside the failing cells, the largest per-configuration seed standard deviation across the ten-seed grid is 0.0085 AUROC, so $\varepsilon = 0.02$ is 2–4$\times$ the variation of re-running the same configuration with a new seed.

**Cross-region uncertainty and reproduction.** Cross-region uncertainty is seed-paired over ten seeds, with each seed contributing its mean over the directed-transfer subset. The 10 primary directional comparisons (five regions with per-chip uncertainty $\times$ two claim families; Cambodia is excluded because its per-chip CIs were not recomputed) (per-region $\text{RF}_{\text{proxy}}-\text{RF}_{\text{true}}$ on F1, and best-FM$-\text{RF}_{\text{true}}$ on AUROC) survive both Benjamini–Hochberg FDR control at 0.05 and Holm correction at 0.05, with minimum $z{=}6.54$ (`ftw_multiple_comparison.json`); chip-paired Wilcoxon signed-rank confirms the India Prithvi-over-U-Net exception ($p{=}2.4 \times 10^{-4}$, all 12 chunks positive). The evaluation code, metric artifacts, and reproduction commands are released; every primary number was independently recomputed from raw rasters and reproduced to three decimals (Appendix A).

## 4 Two Evaluation Choices that Change the Result

All GeoFM entries in this section are the frozen-probe configuration (Prithvi linear probe on extracted per-pixel features) unless labeled 'FT'; see Table 1. Table 2 reports, per region, the per-pixel RF and the strong models under both labels (TRUE-label AUROC unless noted).

Table 2: **Two evaluation choices (true-label AUROC; six regions).** The WorldCover proxy lifts the per-pixel RF from 0.55–0.82 to 0.79–0.96, making it foundation-model-competitive. A from-scratch U-Net on TRUE labels reaches 0.89–0.98 everywhere and is competitive with Prithvi frozen-probe in five of six regions (India excepted), so the reported "GeoFM ≫ baseline" gap mostly reflects the choice of baseline. Prithvi frozen-probe is now reported as a single consistent model (linear probe on Prithvi-EO-2.0 features); a per-model breakdown across {Prithvi, TerraMind, Clay, AnySat} is in Appendix Table 7. Bracketed values are 95% CIs: chip-level bootstrap ($B$=1000) for the frozen-probe column (five regions; Cambodia point estimate only); seed-level $t$-interval ($df$=9) for ten-seed U-Net and FT FM. FT FM = fine-tuned Prithvi (end-to-end). Per-model tile-disjoint sensitivity is in Appendix Table 17. The 'Prithvi frozen-probe' column is a linear probe on Prithvi-EO-2.0 features (*not* the frozen-decoder configuration used in Table 3); see Table 1 for the full configuration key.

| Region | RF TRUE | RF PROXY | U-Net | Prithvi frozen-probe | FT-Prithvi |
|---|---|---|---|---|---|
| India | 0.574 | 0.786 | 0.941 [.925, .958] | 0.983 [.979, .985] | 0.985 [.984, .987] |
| Cambodia | 0.550 | 0.900 | 0.948 [.947, .949] | 0.924 (no CI) | 0.949 [.949, .949] |
| Vietnam | 0.643 | 0.911 | 0.964 [.961, .967] | 0.929 [.925, .933] | 0.958 [.958, .959] |
| Kenya | 0.651 | 0.806 | 0.891 [.885, .897] | 0.766 [.718, .813] | 0.771 [.738, .804] |
| France | 0.695 | 0.959 | 0.977 [.977, .978] | 0.961 [.958, .964] | 0.979 [.978, .980] |
| Netherlands | 0.815 | 0.928 | 0.981 [.980, .983] | 0.955 [.951, .958] | 0.958 [.953, .963] |

**Choice 1: label source.** The proxy raises the per-pixel RF by +0.11 to +0.35 AUROC in every region, turning a near-chance field-membership predictor into one that rivals GeoFMs. The reason is that World-Cover "cropland" is spectrally separable while true field membership is a spatial property. FTW-labeled fields cover a small fraction of a scene, while WorldCover calls a majority of it cropland. The proxy therefore reframes the task as "is this managed land?" rather than "is this pixel in a field?" (Appendix).

**Choice 2: baseline strength.** On TRUE labels a supervised U-Net reaches 0.89–0.98 AUROC in all six regions. Compared against a single consistent FM (Prithvi frozen-probe), the U-Net has higher point AUROC in five of six regions (Vietnam, Kenya, France, Netherlands, Cambodia); India is the certified GeoFM exception (Wilcoxon $p$=2.4 × 10$^{-4}$ on per-chip paired chunks). For Vietnam, the U-Net's ten-seed CI [.961, .967] is disjoint from frozen-Prithvi's [.925, .933]; for Cambodia we have only point estimates (no per-chip CI). India is the only region where frozen-Prithvi's CI [.979, .985] is above the U-Net point estimate (0.941). The "GeoFM ≫ baseline" gap therefore reflects, in substantial part, the choice of baseline rather than a transferable representational advantage. A per-pixel RF cannot use spatial context, and a 5 × 5 windowed RF closes only about 0.01 of the gap (with a small drop, not gain, on Kenya; Table 15). The per-region best across {Prithvi, TerraMind, Clay, AnySat} (Appendix Table 7) raises the frozen-FM Kenya and Netherlands cells to 0.853 (AnySat) and 0.961 (AnySat), but tile-disjoint sensitivity (Appendix Table 17) shows the AnySat advantage does not survive a split-design change; on the apples-to-apples Prithvi-only frozen-probe comparison reported above, the U-Net is the stronger model in those two regions. In-region, Prithvi frozen-probe beats the U-Net in only one of six regions (India), where Prithvi leads by roughly 0.04 AUROC, and the direction is supported by a separate chip-paired one-sided Wilcoxon signed-rank test ($n$=12 paired chunks; $p$=2.4 × 10$^{-4}$; all 12 pairs positive). This chunk audit supports India as a robust regional exception rather than single-seed noise. We retain this exception in the analysis rather than dismissing it. Possible contributors include Prithvi's HLS global pretraining (which may include Indian agro-climates in unknown proportion), India's distinctive smallholder spatial layout, and ordinary single-region noise at the between-region scale. Cambodia, with smaller fields than India, is a tie, so field size alone does not explain the pattern. This U-Net comparison is not compute-matched to the GeoFM full-FT runs (150 epochs from-scratch vs. 80 from a pretrained backbone). An epoch-matched U-Net control at 80 epochs is reported in Appendix Table 13. At the matched 80-epoch budget the U-Net still exceeds the Prithvi frozen-probe in five of six regions (India remains the same certified exception), and the 80-epoch means are within 0.01 AUROC of the 150-epoch means in five of six regions; India shows the largest drop (0.941 → 0.911, seed std 0.023), which widens the existing India frozen-probe lead over the U-Net but does not create a new exception. Under wall-clock rather than epochs the asymmetry is small and runs the opposite direction (U-Net 150 ep ≈15

GPU-min per region vs. full-FT 80 ep ≈20 GPU-min per region, Table 6), so "more epochs for the U-Net" is not a compute advantage. The primary frozen-vs-fine-tuning comparison in Section 5 is separately budget-matched by construction (both 80 epochs, identical AdamW schedule and matched seeds; Table 10). We do not have a single confident explanation and recommend caution before generalizing the supervised-baseline result to small-field, multispectral-pretrained settings.

**External corroboration (PANGAEA).** On AI4SmallFarms (Persello et al., 2023) (manually digitized field boundaries), PANGAEA (Marsocci et al., 2024) reports a from-scratch U-Net at 46.3 mIoU, beating every evaluated GeoFM (Prithvi 26.9, DOFA 27.1, Scale-MAE 21.5). Conversely, the Prithvi-EO-2.0 result of +8.1 mIoU over a from-scratch U-Net on multi-temporal crop segmentation (Szwarcman et al., 2024) comes from a crop-class segmentation task rather than a polygon-derived field-extent benchmark. Both evaluation choices appear in the published literature, independent of our evaluation setup.

## 5 Frozen Decoders Often Match Full Fine-tuning In-Region; Cross-Region Depends on Regime

All GeoFM comparisons in this section use the frozen-decoder and full-FT configurations with matched training seeds; see Table 1. Holding the decoder, recipe, labels, and evaluation pixels fixed, Table 3 compares frozen-backbone GeoFM features against full end-to-end fine-tuning in a single-recipe ten-seed grid with ten matched seeds.

Table 3: **Frozen-decoder vs. full fine-tuning, in-region and cross-region (true-label AUROC).** *Left:* frozen backbone + trained decoder vs. full fine-tuning of the same architecture, with ten matched seeds per cell. 95% CIs on $\Delta = \text{frozen} - \text{fullFT}$ use a seed-paired $t$-interval ($df=9$). A dagger marks TOST-equivalence at $\varepsilon=0.02$ AUROC by the conservative criterion that the 95% CI lies fully within $[-0.02, +0.02]$: 9 of 12 cells pass. Kenya rejects equivalence for both GeoFMs with full-FT ahead; TerraMind India is non-equivalent in the frozen-favored direction because the CI extends past +0.02. *Right:* mean AUROC over all 30 directed transfers and the exploratory non-Kenya 20 subset (see Section 5 for provenance). U-Net is strongest on the full matrix; on exploratory non-Kenya transfers, both frozen GeoFMs beat their full-FT counterparts, and TerraMind frozen is statistically indistinguishable from U-Net.

| In-region: frozen decoder vs. full-FT | | | | | |
|---|---|---|---|---|---|
| Model | Region | frozen | full-FT | $\Delta$ | 95% CI |
| Prithvi | India | 0.982 | 0.985 | −0.004 | $[-0.006, -0.002]^\dagger$ |
| Prithvi | Cambodia | 0.947 | 0.949 | −0.002 | $[-0.003, -0.001]^\dagger$ |
| Prithvi | Vietnam | 0.955 | 0.958 | −0.004 | $[-0.004, -0.003]^\dagger$ |
| Prithvi | Kenya | 0.653 | 0.771 | −0.117 | $[-0.176, -0.059]$ |
| Prithvi | France | 0.982 | 0.979 | +0.003 | $[+0.002, +0.004]^\dagger$ |
| Prithvi | Netherlands | 0.971 | 0.958 | +0.013 | $[+0.008, +0.018]^\dagger$ |
| TerraMind | India | 0.968 | 0.948 | +0.020 | $[+0.005, +0.036]$ |
| TerraMind | Cambodia | 0.942 | 0.948 | −0.006 | $[-0.007, -0.005]^\dagger$ |
| TerraMind | Vietnam | 0.951 | 0.956 | −0.004 | $[-0.005, -0.004]^\dagger$ |
| TerraMind | Kenya | 0.613 | 0.710 | −0.097 | $[-0.146, -0.048]$ |
| TerraMind | France | 0.979 | 0.978 | +0.001 | $[+0.000, +0.002]^\dagger$ |
| TerraMind | Netherlands | 0.961 | 0.952 | +0.009 | $[+0.003, +0.015]^\dagger$ |

| Cross-region mean AUROC | | |
|---|---|---|
| Config | all 30 | expl. non-Kenya 20 |
| U-Net (scratch) | **0.826** | **0.842** |
| TerraMind frozen | 0.781 | 0.839 |
| TerraMind full-FT | 0.781 | 0.813 |
| Prithvi full-FT | 0.780 | 0.810 |
| Prithvi frozen | 0.767 | 0.816 |

$^\dagger$ TOST-equivalent within $\pm0.02$ AUROC (seed-paired $t$-interval, df=9, $n=10$ seeds).

Table 4: **In-region IoU complements AUROC** (mean over 10 seeds, threshold $p$=0.5). The IoU pattern favors the U-Net more often than AUROC: U-Net wins outright in five regions (India, Vietnam, Kenya, France, Netherlands), while Full-FT leads in Cambodia. The Kenya entries for the frozen and full-FT configurations remain near all-negative at threshold 0.5. Full AUROC/AP/F1/IoU breakdown in Appendix Table 18.

| Region (positive %) | U-Net IoU | Frozen-decoder IoU | Full-FT IoU |
|---|---|---|---|
| India (∼0.3%) | **0.455** | 0.386 | 0.433 |
| Cambodia (∼23%) | 0.826 | 0.850 | **0.856** |
| Vietnam (∼18%) | **0.854** | 0.839 | 0.847 |
| Kenya (∼0.1%) | **0.086** | 0.000 | 0.005 |
| France (∼11%) | **0.891** | 0.850 | 0.856 |
| Netherlands (∼3.8%) | **0.810** | 0.699 | 0.709 |

**In-region.** With the backbone frozen and only a decoder trained, performance is statistically equivalent to full fine-tuning in nine of twelve region×model cells across the single-recipe ten-seed grid (Table 3, TOST at $\varepsilon$=0.02 AUROC; count stable for $\varepsilon \in [0.02, 0.03]$; no FT-favored non-equivalence besides Kenya at any $\varepsilon \geq$ 0.010; full per-seed breakdown in Appendix Table 8 and sensitivity in Appendix Table 9). Prithvi is equivalent outside Kenya: India, Cambodia, Vietnam, France, and the Netherlands. TerraMind is equivalent in four of six: Cambodia, Vietnam, France, and the Netherlands. The three non-equivalent cells are informative rather than noisy: Kenya favors full-FT for both GeoFMs (Prithvi $\Delta$=−0.117, CI [−0.176, −0.059]; TerraMind $\Delta$=−0.097, CI [−0.146, −0.048]), while TerraMind India favors the frozen decoder in direction but misses equivalence because the CI extends past the +0.02 margin ($\Delta$=+0.020, CI [+0.005, +0.036]). Outside Kenya, the largest fine-tuning advantage is TerraMind Cambodia at 0.006 AUROC, well inside the declared equivalence band. An earlier apparent fine-tuning advantage came from an architecture confound: comparing a fine-tune-plus-decoder against a frozen linear probe. Giving the frozen backbone the same decoder removes the gap in most in-region cells.

**IoU complements AUROC.** For an imbalanced segmentation task (positive rates spanning 0.1%–23%), AUROC alone can obscure threshold behavior. Table 4 reports mean per-region IoU at threshold 0.5: U-Net wins outright in five regions, while full-FT leads only in Cambodia. The Kenya entries for the frozen and full-FT configurations remain near all-negative at threshold 0.5. Two reading implications follow: (i) the U-Net is the strongest configuration on the segmentation metric more often than the AUROC view suggests (Section 4 Choice 2 gives this interpretation); (ii) the IoU view does not show a systematic full-FT advantage over the frozen decoder. Because these are threshold metrics, we do not make an IoU-equivalence claim.

**What about boundary delineation specifically?** Our task is field-extent segmentation, not boundary delineation or instance separation (Section 3). As a control on the frozen-probe comparisons in Table 2, we re-evaluated the frozen-probe feature classifiers on a boundary zone (within $\leq$ 2 pixels of a field edge, India test set, $n$=203,864 boundary pixels). The five frozen-probe configurations evaluated here (four FMs + per-pixel RF) all fall to near-chance on this slice (Table 5: AUROC 0.57–0.59 for Clay, Prithvi, TerraMind, AnySat, RF), against 0.94–0.98 on the full-region task for the three transformer-based FMs. We did not evaluate the U-Net, frozen-decoder, or full-FT configurations on this slice, so Table 5 supports *only* a frozen-probe scope claim: it does not rank the trained segmentation configurations on boundary pixels. Papers using FTW for true boundary delineation (e.g. object-F1 or Boundary-IoU style metrics) ask a strictly harder question and are outside our scope.

Table 5: **Frozen-probe boundary-zone evaluation (India).** Restricting evaluation to pixels within $\leq 2$ pixels of a field edge ($n$=203,864 pixels) drops the five frozen-probe configurations evaluated here (four FMs + per-pixel RF) to near-chance AUROC, against 0.94–0.98 on the full-region task for the three transformer-based FMs. **Scope.** The U-Net, frozen-decoder, and full-FT configurations were *not* evaluated on this slice; this table supports only a frozen-probe scope claim and does not compare trained segmentation models on boundary pixels. None of the frozen-probe configurations is a boundary-delineation specialist; the frozen-probe field-extent advantage rests on interior context.

| Slice | RF | Clay | Prithvi | TerraMind | AnySat | full-region AUROC reference |
|---|---|---|---|---|---|---|
| boundary zone ($\leq 2$ px) | 0.586 | 0.571 | 0.576 | 0.584 | 0.582 | — |
| full region (India) | 0.574 | 0.577 | 0.983 | 0.967 | 0.939 | cf. Table 2 |

**Cross-region.** Transferring across countries without target-region labels, the ten-seed grid covers all 30 directed train-country→test-country transfers among the six regions (Appendix Table 11). The full matrix does *not* support a broad frozen-GeoFM transfer win: the from-scratch U-Net is strongest overall (0.826 mean AUROC), followed by TerraMind full-FT and TerraMind frozen (both 0.781), Prithvi full-FT (0.780), and Prithvi frozen (0.767). Seed-paired deltas over the 30 transfers show U-Net ahead of both frozen GeoFMs (Prithvi frozen minus U-Net = $-0.059$, CI $[-0.067, -0.051]$; TerraMind frozen minus U-Net = $-0.045$, CI $[-0.053, -0.036]$). Prithvi frozen also trails Prithvi full-FT on the full matrix ($-0.013$, CI $[-0.021, -0.006]$), while TerraMind frozen and TerraMind full-FT are tied ($-0.000$, CI $[-0.013, +0.013]$).

The non-Kenya partition was *not pre-registered*: it was defined after we observed Kenya's in-region full-FT dominance and its threshold-metric collapse. We therefore label subset-derived conclusions *exploratory*; the full 30-transfer matrix remains the confirmatory analysis, and it is under the full matrix that we make the U-Net-is-strongest claim. The frozen-over-FT cross-region result appears in the exploratory non-Kenya subset. Kenya is independently characterized as a sparse-positive case on three grounds before its transfers are partitioned out: extreme label sparsity ($\sim$0.1% positive pixels, the lowest in the study), 155 training chips, and threshold-metric collapse on trained GeoFM configurations (Section 6, Appendix Table 18). On the exploratory non-Kenya 20 directed transfers, both frozen GeoFMs beat their full-FT counterparts: Prithvi frozen minus Prithvi full-FT is $+0.006$ AUROC (CI $[+0.001, +0.011]$), and TerraMind frozen minus TerraMind full-FT is $+0.025$ (CI $[+0.019, +0.032]$). Relative to the U-Net, TerraMind frozen is statistically indistinguishable on exploratory non-Kenya transfers ($-0.004$, CI $[-0.013, +0.005]$), while Prithvi frozen is lower ($-0.026$, CI $[-0.033, -0.019]$). The result is therefore conditional: frozen decoders beat full-FT in-region and on the exploratory non-Kenya subset, but the confirmatory finding on the full 30-transfer matrix is that a from-scratch U-Net remains the strongest cross-region model. The AUROC ordering is preserved under AP. Under IoU@0.5 the ranking is metric-dependent: TerraMind frozen slightly exceeds the U-Net on both subsets ($+0.02$ on all-30, $+0.05$ on the exploratory non-Kenya-20), so the U-Net-over-frozen-GeoFMs statement is scoped to AUROC and AP (Appendix Table 12); on the segmentation-quality metric the frozen GeoFMs are competitive-to-better. This metric-dependence is consistent with the paper's frozen-decoder-often-matches-fine-tuning thesis and does not affect the audit's core point that a task-trained supervised model far exceeds a per-pixel spectral baseline (RF, Table 2).

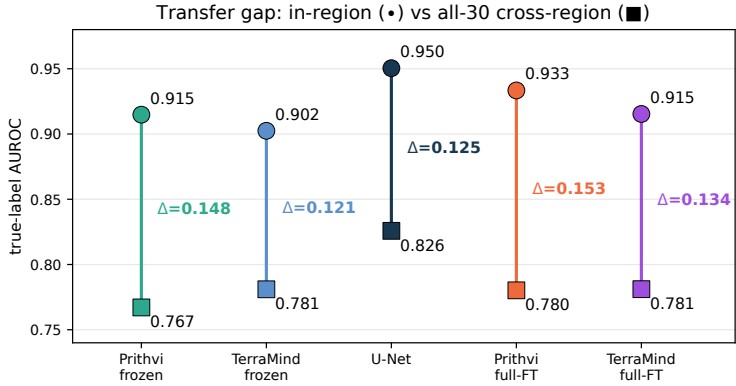

Figure 1: **Transfer gap.** Six-region in-region mean (●) vs. full 30-transfer cross-region mean (■). The U-Net is strongest cross-region on the full matrix; frozen decoders are primarily useful as lower-cost alternatives to full fine-tuning rather than as general cross-region winners.

## 6 Characterizing the Kenya Sparse-Positive Regime

The results in this section use the same frozen-decoder and full-FT configurations as Section 5; see Table 1. Kenya is the consistent sparse-positive boundary case. In-region, full fine-tuning beats the frozen decoder for both GeoFMs in the ten-seed grid: Prithvi frozen 0.653 vs. full-FT 0.771 ($\Delta = -0.117$, CI $[-0.176, -0.059]$), and TerraMind frozen 0.613 vs. full-FT 0.710 ($\Delta = -0.097$, CI $[-0.146, -0.048]$). Cross-region, Kenya-involved transfers are also the hardest subset: on the 10 Kenya-involved directions, U-Net averages 0.793 AUROC, while frozen Prithvi and frozen TerraMind average 0.669 and 0.666. Two measured properties distinguish Kenya. First, label sparsity: Kenya's true-field-pixel coverage in training masks is the lowest of any region at about 0.1%, versus India 0.3%, Netherlands 3.8%, France 11%, Vietnam 18%, Cambodia 23%. This is not field size: Cambodia has the smallest fields (79% under 0.3 ha) but the highest coverage (23%) and does not show the same failure, so parcel size alone does not explain the boundary. Second, training-set size: Kenya has 155 training chips (Cambodia has 107 and does not show the same failure), so training-set size alone is not the mechanism. Spatial fragmentation may also contribute, but we did not quantify it independently. Under this measured combination, frozen-feature decoders are unstable (seed std up to ±0.04), and the supervised U-Net is the most robust tested model on threshold metrics.

**Reconciling Kenya numbers across tables.** Kenya is the one region where the three configurations defined in Section 3 disagree numerically. **Best frozen-probe** (linear classifier on AnySat features, reported in Appendix Table 7 from a separate frozen-probe sensitivity, not the ten-seed decoder grid) scores 0.853 AUROC; **frozen-decoder** (Prithvi features, 2.75M-parameter decoder, Table 3) scores 0.653; the ten-seed threshold-metrics mean (Appendix Table 18) is 0.653 AUROC and 0.000 IoU. Tile-disjoint sensitivity (Appendix Table 17) further shows that AnySat's chip-grouped Kenya score does not survive a split-design change (0.853 → 0.580), so this best-of-four Appendix cell is fragile in addition to being from a different configuration. The pattern is consistent with, but does not prove, a *feature-adaptability* explanation: under ∼0.1% positive pixels and only 155 training chips, the discriminative signal sits in feature dimensions a generic pretraining objective did not prioritize, and a frozen backbone cannot reshape its representation to surface them; full fine-tuning (307M trainable parameters) and a from-scratch U-Net (31M) both *can* reshape features and both outscore the frozen-decoder configuration (0.771 and 0.891 vs 0.653). This observation argues against a simple explanation based only on the decoder having too many parameters, but it does not identify the causal mechanism. In Section 7 and the cross-region narrative, we use the frozen-decoder Prithvi number (0.653) as the canonical Kenya frozen value, since it is the apples-to-apples comparator against fine-tuning. Threshold metrics sharpen the boundary: at probability threshold 0.5, the frozen decoder predicts all negatives and full fine-tuning is near all-negative on Kenya (mean IoU 0.000 and 0.005; Appendix Table 18); only the supervised U-Net recovers substantial positives (mean IoU 0.086). Kenya is a boundary case across both rank and threshold metrics, not an AUROC artifact.

We do not state a hard numeric threshold. With one region below about 0.2% coverage, the transition is an $n=1$ observation, and a precise cutoff would overclaim. A working diagnostic instead: when annotated positives are very sparse ($\lesssim$0.2% of pixels) and the training set is small, treat frozen-feature decoders as a risk and compare them directly with a from-scratch supervised CNN. Practitioners can check their own positive rate and seed variance to decide. Establishing the exact boundary is future work.

Appendix Table 14 places Kenya in the context of the six regions. Extreme positive-pixel sparsity ($\sim 0.1\%$) and small training set (155 chips) co-occur in Kenya alone, so with only six regions we do not attempt a formal sparsity threshold. Two clean directional statements are supportable from existing artifacts. First, sample-size alone is not the mechanism: subsampling dense-positive Cambodia full-fine-tuning to 11, 27, and 54 training chips (far below Kenya's 155) leaves AUROC essentially flat (0.940–0.949 vs. 0.949 at the full 107 chips; Appendix; data in `ftw_finetune_fm_prithvi_camld_*.json`), and threshold metrics do not collapse in the way they do in Kenya. Second, the direction of the frozen-vs-fine-tuning $\Delta$AUROC is not monotone in positive-%: Netherlands ($\sim$3.8% positives) is frozen-favored, France ($\sim$11%) and Vietnam ($\sim$18%) are ties, and Cambodia ($\sim$23%) is a small FT-lead in the equivalence band. The Kenya failure is therefore best characterized as the joint sparse-positive + small-training-set regime rather than either factor alone; separating them with a controlled frozen-decoder low-data arm at Kenya-scale chip count is future work. Specifically, the reverse arm (subsampling a dense-positive region such as Cambodia at $\sim$23% positives down to a Kenya-level $\sim$0.1% positive rate while holding chip count fixed) would isolate positive-rate from sample-size and chip count, and is the missing controlled arm we register here.

## 7  Cost-Efficiency of Frozen Features

Table 6 contrasts trainable parameters and per-region training cost. A frozen backbone with a decoder trains 2.75M parameters versus 307M for full fine-tuning, about 110$\times$ fewer, while matching AUROC in the 9 of 12 TOST-equivalent region$\times$model cells (Table 8) and placing TerraMind India essentially at the equivalence margin by point estimate ($\Delta$=+0.020) but not by CI. Kenya is the only region where full-FT decisively wins for both GeoFMs. A frozen linear probe trains about $10^3$ parameters in seconds on CPU. AUROC parity with the 31M-parameter U-Net is region-dependent: in the ten-seed in-region decoder comparison, U-Net has higher point AUROC than frozen-Prithvi in four of six regions (Cambodia, Vietnam, Kenya, Netherlands), while frozen-Prithvi is higher in India and narrowly in France. The largest gap is Kenya, where U-Net leads frozen-Prithvi by 0.237 AUROC. On IoU, the U-Net wins outright outside Cambodia (Table 4). The frozen-probe cost advantage is robust; matched-accuracy equality vs the supervised baseline is region-specific. For cross-region deployment, the frozen decoder also needs no target-region labels, but the full transfer matrix shows that it should be compared directly with a supervised U-Net rather than assumed to transfer best.

Table 6: **Cost asymmetry.** Trainable parameters and per-region training cost. The frozen-decoder pipeline is TOST-equivalent to full fine-tuning in 9 of 12 region$\times$model cells (Table 8) and within 0.02 AUROC in one more by rounded point estimate, at a small fraction of the trainable parameters. AUROC parity with the from-scratch U-Net is region-dependent; on IoU, the U-Net wins outright outside Cambodia (Table 4).

| Method | trainable params | per-region train | target labels |
|---|---|---|---|
| per-pixel RF | N/A (fitted trees) | seconds, CPU | yes |
| frozen FM + linear probe | $\sim$1,000 | seconds, CPU | yes (source only, transfer) |
| frozen backbone + decoder | 2.75M | $\sim$12 GPU-min | yes |
| supervised U-Net | 31M | $\sim$15 GPU-min | yes |
| full fine-tune (Prithvi) | 307M | $\sim$20 GPU-min | yes |

**Cost accounting: what is and is not counted.** Table 6 reports trainable parameters and observed per-region training wall-clock on a single A6000 GPU. The frozen-decoder advantage we rely on is precisely scoped to two facts. First, it trains $\sim$110$\times$ fewer parameters than full fine-tuning (2.75M decoder parameters vs. 307M Prithvi full-FT parameters). Second, at inference the frozen-decoder and full-FT configurations run the identical backbone+decoder graph, so per-scene inference cost is equal; frozen does not reduce

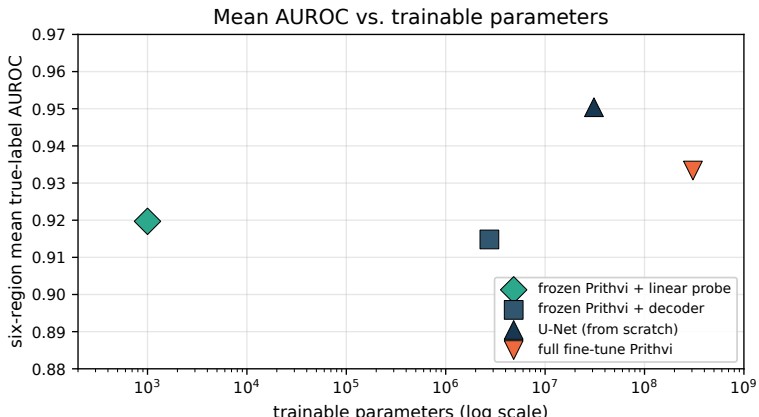

Figure 2: **Cost vs. performance.** Six-region mean in-region AUROC against trainable parameters (log scale). All GeoFM points use Prithvi consistently. Frozen options (a ∼1K-parameter linear probe or a 2.75M-parameter decoder) sit far left of the fully trained models. The frozen *decoder* is TOST-equivalent to the 307M full fine-tune in 9 of 12 region×model cells (Table 8). The frozen *probe* has lower point AUROC than the 31M from-scratch U-Net in most regions; India is the main GeoFM exception.

deployment-time backbone compute. The released frozen-decoder pipeline does not cache features offline: the frozen backbone still runs a forward pass every training epoch, so there is no per-epoch compute saving from cached features and no cached-feature storage term to report. A user who chooses to cache frozen features offline would pay a one-time extraction and storage cost and could then reuse those features across multiple task heads, but our released pipeline does not do this and therefore does not realize that multi-head-reuse saving.

## 8 A Corrected-Evaluation Protocol

We distill the study into a ten-point checklist for field-level GeoFM benchmarking. **Labels:** (1) evaluate against field-extent labels derived from field polygons, not a land-cover proxy; treat proxy-only rankings as tentative. (2) Make the label comparison same-pixel paired. **Baselines:** (3) include a strong task-specific supervised baseline (U-Net), not only a per-pixel spectral model. (4) Report a frozen-feature probe and a fine-tuned variant; do not assume fine-tuning helps. **Splits:** (5) group by chip. (6) Add a scene or tile-disjoint split, and a cross-region split. **Preprocessing:** (7) use each model's own normalization; exclude no-data. **Statistics:** (8) cluster uncertainty at the chip level, use multiple seeds, apply multiple-comparison correction. **Interpretation:** (9) run an edge-vs-interior control before claiming boundary understanding. (10) Report across field-size and label-sparsity regimes, and report trainable-parameter and compute cost alongside accuracy.

**Worked example: re-reading a published ranking.** The Prithvi-EO-2.0 report (Szwarcman et al., 2024) reports a +8.1 mIoU gap for Prithvi-EO-2.0-600M over a from-scratch U-Net on multi-temporal crop segmentation (50.7 vs. 42.6 mIoU in its Table 9). Protocol step (1) flags direct transfer of that ranking to our setting as task-dependent: crop-class segmentation is not polygon-derived field-extent segmentation, so the result should be re-tested on field-polygon labels before being treated as evidence about field membership. With polygon-derived labels and a strong baseline (steps 1 to 3), PANGAEA (Marsocci et al., 2024) reports on AI4SmallFarms (Persello et al., 2023) (manually digitized parcels) a from-scratch U-Net at 46.3 mIoU, beating every evaluated GeoFM (Prithvi 26.9, DOFA 27.1, Scale-MAE 21.5). Multi-temporal crop segmentation and single-date AI4SmallFarms parcel delineation are different tasks/datasets, so this is not a direct apples-to-apples inversion of the Prithvi result; rather, it illustrates why proxy-label wins should be re-tested under steps (1)–(3).

**Practitioner implications:** (i) Cost: a frozen backbone with a decoder trains about $110\times$ fewer trainable parameters than full fine-tuning, and a linear probe about $10^3$ (Table 6). The released frozen-decoder pipeline does not cache features: it still runs the backbone forward pass during training, and it uses the same backbone+decoder graph as full-FT at inference. Users who explicitly cache features offline can trade one-time extraction and storage for multi-head reuse, but that saving is not realized in the reported runs. (ii) Label-scarce deployment: for a new region with no local labels, the full cross-region matrix supports frozen decoders over full fine-tuning on the exploratory non-Kenya transfer subset for both GeoFMs, but it does not support a broad claim that frozen GeoFMs beat a source-trained U-Net. On the full 30-transfer matrix the U-Net is strongest; on exploratory non-Kenya transfers TerraMind frozen is statistically tied with the U-Net, while Prithvi frozen is lower. (iii) Model selection: for this single-date task, a frozen decoder is a low-cost configuration to test first, not a general replacement for full fine-tuning or a substitute for a strong supervised baseline; sparse-positive regimes require a direct comparison with a supervised CNN.

## 9    Discussion, Limitations, Conclusion

**Scope.** Our claims are scoped to agricultural field-extent segmentation from single-date Sentinel-2 with true field labels (per-pixel inside-vs-outside field polygon; we do not evaluate boundary delineation or instance separation). They do not necessarily extend to other tasks (e.g. crop-type, change detection), other sensors, or multi-temporal regimes. GeoFM pretraining may help there.

**Broader impact.** Automatic field-extent maps are used downstream in agricultural planning, subsidy allocation, crop monitoring, and land-tenure assessment. The same maps can also enable land-use surveillance, regulatory enforcement, or asset valuation that affects smallholder farmers without their consent. Our methodology choices (true labels rather than land-cover proxies, a strong supervised baseline, explicit reporting of boundary-zone failure) aim to keep claims about "what works" aligned with the actual task and to make weakness in sparse-label regimes (e.g. Kenya) explicit. Deployers should not generalize results across agro-climatic regimes without re-evaluating on local labels, and decisions with consequences for individuals should not be automated from these maps alone.

**Scope of the claim.** The claim is not that GeoFMs are useless. Current evaluations overstate their advantage for in-distribution field mapping with reasonable supervision; the practical value here is as frozen feature extractors, which are low-cost and often match full fine-tuning in-region. Cross-region, the value is narrower: frozen decoders beat full fine-tuning on the exploratory non-Kenya transfer subset, but the U-Net is strongest on the full directed matrix.

**Limitations.** Four GeoFMs evaluated here (PANGAEA covers more, with a similar supervised-baseline pattern); a single supervised architecture (PANGAEA provides corroborating U-Net results); FTW partial annotation in sparse regions (our three-region sensitivity check does not indicate that it drives the frozen-FM result, Appendix); and cross-region transfer is unfavorable to frozen GeoFMs on the full 30-transfer matrix relative to the U-Net, with the frozen-over-FT result limited to the exploratory non-Kenya subset. The U-Net-vs-GeoFM comparisons are architecture-appropriate but not compute-matched: the U-Net uses Adam for 150 epochs from random initialization while the GeoFM frozen-decoder and full-FT runs use AdamW for 80 epochs from pretrained weights. An 80-epoch U-Net control (Appendix Table 13, 3 seeds per region) preserves the qualitative pattern of Table 2 within 0.01 AUROC in five of six regions; India shows a 0.03 drop that widens the existing India frozen-probe lead but does not create a new region-level exception. The frozen-decoder-vs-full-FT primary comparison is separately budget-matched by construction (Table 10). Under wall-clock, the U-Net is cheaper than full-FT per region despite its longer schedule, so the epoch mismatch is not a compute advantage for the U-Net. A per-epoch convergence diagnostic (Appendix Figure 3) shows the Vietnam frozen-decoder curve plateaus by epoch 15 and drifts within 0.002 AUROC through epoch 90; the Vietnam full-FT curve peaks near epoch 90 and slightly declines by epoch 150; and the Kenya full-FT curve peaks at epoch 15 and declines to 0.702 by epoch 150. The 80-epoch primary comparison is therefore not under-trained for the frozen-vs-fine-tuning finding; for Kenya specifically, longer training strictly hurts full-FT. The boundary-zone control (Table 5) is limited to the frozen-probe configurations; extending it to the U-Net, frozen-decoder, and full-FT configurations would require regenerating per-pixel predictions for those

trained models. We therefore report Table 5 strictly as a frozen-probe scope check. Feature-extraction wall-time is not separately tabulated because the frozen-decoder configuration used in the frozen-vs-fine-tuning spine does not cache features to disk; extraction is folded into the training wall-time (Table 6).

**Connection to the broader foundation-model literature.** For geospatial models, the cross-region result is partly consistent with what Kumar et al. (2022) showed for vision: fine-tuning can distort pretrained features and underperform frozen probing under distribution shift. We see that pattern on the exploratory non-Kenya transfer subset, where frozen decoders beat full fine-tuning for both GeoFMs. The full 30-transfer matrix is more cautious because Kenya-involved transfers favor task-trained supervised models and the U-Net is strongest overall. This supports a narrower recommendation: test a frozen decoder before full fine-tuning, but do not treat it as a transfer rule. The released evaluation setup can test where this pattern persists.

**Conclusion.** For single-date Sentinel-2 field-extent segmentation, our controlled six-region study supports an audit-first reading. First, label choice matters: replacing polygon-derived field-extent labels with a land-cover proxy lifts a per-pixel RF into the GeoFM-competitive range while changing the task. Second, baseline choice matters: a from-scratch U-Net on true labels reaches 0.89–0.98 AUROC and has higher point AUROC than the Prithvi frozen-probe outside India, which remains the frozen-probe exception. These two choices should be audited before interpreting a GeoFM ranking as evidence of transferable representation quality.

Under that corrected evaluation, the fine-tuning answer is more conditional than the published default suggests. In a single-recipe 10-seed analysis, frozen-decoder features are TOST-equivalent to full fine-tuning in 9 of 12 region×model cells (Prithvi equivalent outside Kenya; TerraMind equivalent in Cambodia, Vietnam, France, and the Netherlands). Kenya is the documented exception (label sparsity ∼0.1%): full-FT decisively wins for both GeoFMs. TerraMind India is the remaining non-equivalent cell, in the frozen-favored direction but with CI past the equivalence margin. The result is established for this single-date setting; multi-temporal regimes (where Prithvi-EO-2.0 was designed to operate) are out of scope.

Cross-region, the conclusion is conditional rather than uniformly frozen-favored. On the full 30 directed-transfer matrix, a from-scratch U-Net is the strongest configuration, and both frozen GeoFMs trail it. On the exploratory non-Kenya 20-transfer subset, both frozen GeoFMs beat their full-FT counterparts by seed-paired CIs excluding zero, and TerraMind frozen is statistically tied with the U-Net. Thus the practical recommendation is to test a frozen decoder as a low-cost alternative to full fine-tuning, but to keep a strong supervised U-Net in the benchmark before claiming cross-region transfer advantages. The equivalent frozen-decoder cells use roughly 110× fewer trainable parameters than full fine-tuning. We release the code, metrics, protocol, and artifacts so that future GeoFM evaluations can audit both evaluation choices directly.

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

## A Reproducibility and Controls

This appendix collects the reproducibility details and sensitivity controls referenced from the main text: the per-model frozen-probe breakdown (Table 7), seed-paired TOST detail (Table 8) and its $\varepsilon$-sensitivity sweep (Table 9), the training-recipe key (Table 10), the full cross-region transfer matrix (Table 11) and its AP/IoU companion (Table 12), U-Net epoch sensitivity (Table 13) and Kenya convergence trajectories (Figure 3), the per-region operational regime (Table 14), the spatial-pooling control (Table 15), the FTW partial-labeling sensitivity (Table 16), the tile-disjoint split sensitivity (Table 17), and the full threshold-metric breakdown (Table 18). All numbers are recomputed from the released artifact.

Table 7: **Per-region breakdown of frozen-probe AUROC across four FMs** (true-label, linear probe, deterministic given the split). The Table 2 main column uses Prithvi-EO-2.0 consistently; this table reports each FM separately for transparency. **Bold** marks the per-region winner. AnySat wins Kenya (0.853) and Netherlands (0.961), but tile-disjoint sensitivity (Table 17) shows the AnySat scores do not survive a split-design change (0.853 → 0.580 on Kenya; 0.943 → 0.605 on France); we therefore use the Prithvi-only column as the main comparison. Bootstrap 95% CIs from $B$=1000 chip-level resamples.

| Region | Prithvi | TerraMind | Clay | AnySat |
|---|---|---|---|---|
| India | **0.983** [.979,.985] | 0.967 [.963,.971] | 0.577 [.554,.601] | 0.939 [.915,.959] |
| Vietnam | **0.929** [.925,.933] | 0.916 [.911,.921] | 0.503 [.495,.511] | 0.897 [.880,.915] |
| Kenya | 0.766 [.718,.813] | 0.745 [.699,.789] | 0.556 [.505,.599] | **0.853** [.825,.881] |
| France | **0.961** [.958,.964] | 0.951 [.947,.954] | 0.501 [.494,.509] | 0.943 [.927,.957] |
| Netherlands | 0.955 [.951,.958] | 0.943 [.938,.948] | 0.498 [.488,.507] | **0.961** [.956,.967] |

Cambodia per-chip bootstrap CIs were not recomputed; available per-model point estimates are Prithvi 0.924, TerraMind 0.918, and Clay 0.491. No Cambodia AnySat result is present in the artifact.

Table 8: **Seed-paired TOST equivalence for frozen-decoder vs. full fine-tune (12 cells: two Ge-oFMs × six regions).** Per-seed AUROC values (seeds 0–9), paired delta $\Delta_s = \text{frozen}_s - \text{FT}_s$ averaged over seeds, 95% paired $t$-interval at df=9, and TOST verdict at declared equivalence margin $\varepsilon$=0.02. The paired analysis uses matched random seeds for each frozen/FT pair in the single-recipe 10-seed grid. Source files and aggregate JSON are generated by `scripts/integrate_headline_results.py`. Total TOST-equivalent: 9 of 12.

| Model | Region | frozen seeds (s0–s9) | FT seeds (s0–s9) | paired $\Delta \pm \sigma$ | 95% CI |
|---|---|---|---|---|---|
| Prithvi | India | .984, .981, .983, .985, .983, .979, .980, .979, .985, .978 | .982, .988, .985, .986, .986, .986, .983, .982, .988, .986 | $-0.004 \pm 0.003$ | $[-0.006, -0.002]$ ✓ |
| Prithvi | Cambodia | .948, .948, .946, .947, .946, .946, .948, .948, .948, .947 | .949, .949, .948, .950, .949, .949, .949, .949, .949, .948 | $-0.002 \pm 0.001$ | $[-0.003, -0.001]$ ✓ |
| Prithvi | Vietnam | .954, .955, .954, .955, .955, .955, .955, .954, .955, .955 | .958, .957, .958, .959, .959, .959, .958, .958, .958, .960 | $-0.004 \pm 0.001$ | $[-0.004, -0.003]$ ✓ |
| Prithvi | Kenya | .667, .658, .615, .599, .607, .676, .531, .723, .745, .713 | .781, .798, .699, .811, .814, .728, .761, .701, .791, .824 | $-0.117 \pm 0.082$ | $[-0.176, -0.059]$ × |
| Prithvi | France | .981, .982, .981, .982, .982, .982, .982, .981, .981, .981 | .980, .978, .981, .979, .979, .977, .977, .979, .979, .979 | $+0.003 \pm 0.002$ | $[+0.002, +0.004]$ ✓ |
| Prithvi | Netherlands | .971, .973, .971, .971, .970, .974, .971, .968, .968, .971 | .964, .957, .953, .954, .963, .961, .946, .964, .951, .965 | $+0.013 \pm 0.007$ | $[+0.008, +0.018]$ ✓ |
| TerraMind | India | .972, .934, .976, .977, .973, .975, .978, .964, .970, .965 | .970, .955, .950, .946, .961, .956, .946, .930, .912, .955 | $+0.020 \pm 0.021$ | $[+0.005, +0.036]$ × |
| TerraMind | Cambodia | .943, .942, .943, .942, .941, .943, .943, .942, .942, .942 | .949, .948, .947, .949, .949, .948, .949, .949, .949, .947 | $-0.006 \pm 0.001$ | $[-0.007, -0.005]$ ✓ |
| TerraMind | Vietnam | .951, .951, .952, .951, .951, .951, .952, .951, .951, .951 | .955, .956, .956, .956, .956, .956, .956, .956, .955, .956 | $-0.004 \pm 0.000$ | $[-0.005, -0.004]$ ✓ |
| TerraMind | Kenya | .610, .632, .638, .596, .537, .576, .707, .643, .569, .625 | .820, .724, .788, .611, .573, .696, .696, .773, .721, .703 | $-0.097 \pm 0.069$ | $[-0.146, -0.048]$ × |
| TerraMind | France | .978, .979, .979, .979, .979, .979, .979, .979, .979, .978 | .979, .978, .978, .978, .977, .979, .975, .978, .978, .978 | $+0.001 \pm 0.001$ | $[+0.000, +0.002]$ ✓ |
| TerraMind | Netherlands | .960, .958, .958, .956, .964, .965, .964, .960, .964, .959 | .942, .951, .961, .953, .943, .964, .962, .940, .954, .952 | $+0.009 \pm 0.009$ | $[+0.003, +0.015]$ ✓ |

✓ = TOST equivalent at $\varepsilon$=0.02 AUROC (95% paired CI fully within ±0.02; equivalent to standard TOST at $\alpha$=0.025 per side, conservative form). × = not formally equivalent. Kenya is a full-FT win for both GeoFMs; TerraMind India is frozen-favored but its CI extends past +0.02.

Table 9: **Equivalence sensitivity to the TOST margin.** Counts of in-region region×model cells declared equivalent under seed-paired TOST at $df$=9, for six values of $\varepsilon$ (AUROC). The failing set is invariant across $\varepsilon \in [0.020, 0.030]$. Cells added at stricter margins in $[0.010, 0.020)$ are frozen-favored (F), so tightening the margin in this range only reclassifies cells where the frozen decoder already beats fine-tuning. At $\varepsilon$=0.005 two additional cells become formally non-equivalent in the FT-favored direction (T), but their $\Delta$ magnitudes (0.004 and 0.006) are below the operational tolerance motivating the margin choice.

| $\varepsilon$ | Equivalent | Newly failing (direction, $\Delta$) | Failing cells at this $\varepsilon$ |
|---|---|---|---|
| 0.030 | 9/12 | — | {Pr–Kenya (T), TM–India (F), TM–Kenya (T)} |
| 0.025 | 9/12 | — | {same} |
| 0.020 | 9/12 | — | {same} |
| 0.015 | 8/12 | Pr–Netherlands (F, +0.013) | {above} ∪ Pr–Netherlands |
| 0.010 | 7/12 | TM–Netherlands (F, +0.009) | {above} ∪ TM–Netherlands |
| 0.005 | 5/12 | Pr–India (T, −0.004); TM–Cambodia (T, −0.006) | {above} ∪ Pr–India, TM–Cambodia |

All experiments use the same canonical split (chip-grouped, seed 20260514, overlap 0). Primary numbers were independently recomputed from raw rasters and features and reproduced to three decimals; label-to-imagery alignment was verified at 100% (every positive pixel inside a field polygon, every negative outside) in all six regions. Controls: a per-pixel ablation ($1 \times 1$ convolutions) falls to $\approx$RF, and a $5 \times 5$ windowed RF gains only about 0.01 AUROC on average (Kenya is the exception, dropping 0.010), which confirms that the U-Net and GeoFM advantage is learned spatial context rather than naive pooling. An edge-pixel control localizes the advantage to within-field interior context. An FTW partial-labeling sensitivity test (restricting negatives to high-confidence non-cropland) leaves the GeoFM true-task AUROC almost unchanged (e.g. India Prithvi 0.983 → 0.987). The single-recipe ten-seed grid covers all 30 directed cross-region pairs for U-Net, Prithvi frozen/full-FT, and TerraMind frozen/full-FT; derived summaries are generated by `scripts/integrate_headline_results.py`. Legacy pilot-run result files remain under `data/results/archive/historical_iterations/` for traceability but are not used for the main-text claims. Code, the evaluation setup, and all metric artifacts (with checksums and end-to-end reproduction commands) are included in the supplementary material submitted alongside this paper (anonymized for

double-blind review; Croissant metadata are included, and a persistent DOI is planned for the camera-ready).

Table 10: **Training recipes and controlled factors.** Frozen-decoder and full-FT GeoFM runs share the same decoder, loss, augmentation, schedule, and matched seeds; they differ only in whether the backbone is updated. The U-Net uses its architecture-appropriate optimizer and longer from-scratch schedule. Evaluation pixels, labels, and the chip-grouped split are identical for every model and seed. The U-Net and GeoFM training budgets differ by design (150 ep from-scratch vs. 80 ep from pretrained), matching common practice for each architecture. An 80-epoch U-Net sensitivity (Appendix Table 13) and a per-epoch convergence diagnostic (Appendix Figure 3) are provided for audit purposes.

| Component | Setting |
|---|---|
| Loss | BCE (capped positive weight, $\leq 50$) + Dice |
| Optimizer | U-Net: Adam, lr $10^{-3}$; GeoFM: AdamW (backbone $10^{-4}$, decoder $10^{-3}$) |
| LR schedule | linear warmup (5 epochs, start factor 0.1) + cosine decay |
| Augmentation | random horizontal/vertical flips |
| Input | U-Net: native chip, 12 S2 bands (reflect-padded); GeoFM: 224×224 |
| Decoder | GeoFM: shared 4-stage bilinear-upsample conv decoder (identical for frozen & fine-tuned); U-Net: 4-level (31M) |
| Epochs | U-Net 150 (from scratch); GeoFM full-FT / frozen-decoder 80 |
| Selection | last epoch (no early stopping); mean±std over 10 matched training seeds in the ten-seed grid |

Table 11: **Full cross-region transfer matrix** (true-label AUROC, mean over 10 seeds; train country A, test country B's canonical test pixels). The table covers all 30 directed transfers among the six regions. U-Net is strongest on the full matrix (0.826 mean). On the exploratory non-Kenya 20 subset, TerraMind frozen (0.839) is close to U-Net (0.842) and beats TerraMind full-FT (0.813), while Prithvi frozen (0.816) beats Prithvi full-FT (0.810) but remains below U-Net. Seed-paired CIs for subset-level deltas are generated in `data/results/ftw_cross_region_transfer.json`.

| A→B | U-Net | Prithvi-FT | Prithvi-frz | TerraMind-FT | TerraMind-frz |
|---|---|---|---|---|---|
| India→Cambodia | 0.719 | 0.646 | 0.629 | 0.627 | 0.697 |
| India→Vietnam | 0.698 | 0.664 | 0.653 | 0.643 | 0.699 |
| India→Kenya | 0.696 | 0.629 | 0.579 | 0.546 | 0.616 |
| India→France | 0.824 | 0.750 | 0.644 | 0.611 | 0.660 |
| India→Netherlands | 0.830 | 0.746 | 0.675 | 0.614 | 0.646 |
| Cambodia→India | 0.832 | 0.861 | 0.890 | 0.870 | 0.885 |
| Cambodia→Vietnam | 0.874 | 0.926 | 0.922 | 0.940 | 0.917 |
| Cambodia→Kenya | 0.881 | 0.836 | 0.865 | 0.862 | 0.853 |
| Cambodia→France | 0.865 | 0.887 | 0.910 | 0.920 | 0.884 |
| Cambodia→Netherlands | 0.895 | 0.883 | 0.884 | 0.901 | 0.879 |
| Vietnam→India | 0.873 | 0.868 | 0.870 | 0.875 | 0.886 |
| Vietnam→Cambodia | 0.939 | 0.938 | 0.938 | 0.945 | 0.930 |
| Vietnam→Kenya | 0.897 | 0.862 | 0.884 | 0.890 | 0.885 |
| Vietnam→France | 0.925 | 0.907 | 0.919 | 0.930 | 0.931 |
| Vietnam→Netherlands | 0.884 | 0.843 | 0.898 | 0.898 | 0.898 |
| Kenya→India | 0.790 | 0.678 | 0.517 | 0.642 | 0.596 |
| Kenya→Cambodia | 0.884 | 0.723 | 0.622 | 0.734 | 0.592 |
| Kenya→Vietnam | 0.746 | 0.642 | 0.588 | 0.690 | 0.543 |
| Kenya→France | 0.800 | 0.767 | 0.629 | 0.692 | 0.550 |
| Kenya→Netherlands | 0.851 | 0.770 | 0.640 | 0.738 | 0.596 |
| France→India | 0.887 | 0.877 | 0.903 | 0.891 | 0.904 |
| France→Cambodia | 0.840 | 0.740 | 0.748 | 0.781 | 0.750 |
| France→Vietnam | 0.835 | 0.688 | 0.742 | 0.794 | 0.828 |
| France→Kenya | 0.761 | 0.672 | 0.722 | 0.714 | 0.735 |
| France→Netherlands | 0.937 | 0.905 | 0.931 | 0.883 | 0.926 |
| Netherlands→India | 0.864 | 0.856 | 0.898 | 0.871 | 0.881 |
| Netherlands→Cambodia | 0.691 | 0.700 | 0.691 | 0.716 | 0.777 |
| Netherlands→Vietnam | 0.692 | 0.619 | 0.627 | 0.648 | 0.834 |
| Netherlands→Kenya | 0.625 | 0.618 | 0.642 | 0.663 | 0.692 |
| Netherlands→France | 0.940 | 0.901 | 0.954 | 0.904 | 0.960 |
| mean all 30 | 0.826 | 0.780 | 0.767 | 0.781 | 0.781 |
| mean expl. non-Kenya 20 | 0.842 | 0.810 | 0.816 | 0.813 | 0.839 |

Table 12: **Cross-region AP and IoU alongside AUROC (mean over 10 seeds per pair, then mean over pairs).** Threshold metrics (IoU at 0.5) and AP are reported in addition to AUROC because low positive rates (Kenya $\sim 0.1\%$, India $\sim 0.3\%$) can make AUROC alone optimistic. The AUROC ordering is preserved under AP for both the full 30 directed transfers and the exploratory non-Kenya 20 subset. The IoU@0.5 column is a thresholded segmentation-quality metric; under this metric the ordering is metric-dependent (see Section 5). Aggregated from existing per-run JSONs; no new experiments. See `data/results/ftw_xfer_metrics.json`.

| Config | all 30 directed transfers | | | exploratory non-Kenya 20 | | |
|---|---|---|---|---|---|---|
| | AUROC | AP | IoU@0.5 | AUROC | AP | IoU@0.5 |
| U-Net (scratch) | **0.826** | **0.733** | 0.335 | **0.842** | 0.805 | 0.423 |
| TerraMind frozen | 0.781 | 0.696 | **0.353** | 0.839 | **0.804** | **0.469** |
| TerraMind full-FT | 0.781 | 0.693 | 0.318 | 0.813 | 0.778 | 0.416 |
| Prithvi full-FT | 0.780 | 0.689 | 0.308 | 0.810 | 0.771 | 0.402 |
| Prithvi frozen | 0.767 | 0.682 | 0.308 | 0.816 | 0.782 | 0.404 |

Table 13: **U-Net epoch sensitivity (mean over 3 matched seeds per region).** The 80-epoch column is an epoch-matched control to the GeoFM training budget (both trained with the same recipe otherwise); the 150-epoch column is the best-effort deployable baseline retained from the main paper. At 80 epochs the U-Net remains within 0.01 AUROC of the 150-epoch mean in five of six regions; India shows a 0.03 drop but the paper's certified India exception (Prithvi frozen-probe leads the U-Net) is unchanged. Training-time cost per region is roughly proportional to epochs (Table 6).

| Region | U-Net@150ep | U-Net@80ep | $\Delta$ AUROC | Prithvi frozen |
|---|---|---|---|---|
| India | 0.941 | $0.911 \pm 0.023$ | $-0.030$ | 0.983 |
| Cambodia | 0.948 | $0.947 \pm 0.001$ | $-0.001$ | 0.924 |
| Vietnam | 0.964 | $0.955 \pm 0.001$ | $-0.009$ | 0.929 |
| Kenya | 0.891 | $0.889 \pm 0.004$ | $-0.002$ | 0.766 |
| France | 0.977 | $0.976 \pm 0.001$ | $-0.001$ | 0.961 |
| Netherlands | 0.981 | $0.982 \pm 0.003$ | $+0.001$ | 0.955 |

Table 14: **Per-region operational regime (descriptive, existing artifacts only).** Positive-pixel prevalence and training-set size vary by orders of magnitude across the six regions. Combined with the frozen-decoder-vs-full-fine-tuning $\Delta$AUROC (Prithvi and TerraMind) and threshold-metric behavior (U-Net vs. frozen-decoder IoU@0.5), this gives a compact view of when frozen features are safe and when they fail. Kenya is the only region below $\sim 0.2\%$ positive pixels, so we do not report a formal sparsity threshold; the pattern is a description, not an estimate. Sample-size alone is not the mechanism: subsampling Cambodia (dense-positive) full-fine-tuning to 11–54 training chips leaves AUROC essentially flat (0.940–0.949; `ftw_finetune_fm_prithvi_camld_*`). A controlled frozen-decoder low-data arm at Kenya-scale chip count is future work.

| Region | pos. % | train chips | $\Delta$AUROC Pr | $\Delta$AUROC TM | U-Net IoU | Frozen IoU | Regime |
|---|---|---|---|---|---|---|---|
| Kenya | $\sim 0.1$ | 155 | $-0.117$ | $-0.097$ | 0.086 | 0.000 | sparse-positive collapse |
| India | $\sim 0.3$ | 525 | $-0.004$ | $+0.020$ | 0.455 | 0.386 | smallholder; AUROC-tied |
| Netherlands | $\sim 3.8$ | 522 | $+0.013$ | $+0.009$ | 0.810 | 0.699 | frozen-favored AUROC |
| France | $\sim 11$ | 532 | $+0.003$ | $+0.001$ | 0.891 | 0.850 | industrial; near-parity |
| Vietnam | $\sim 18$ | 213 | $-0.004$ | $-0.004$ | 0.854 | 0.839 | tie |
| Cambodia | $\sim 23$ | 107 | $-0.002$ | $-0.006$ | 0.826 | 0.850 | tie |

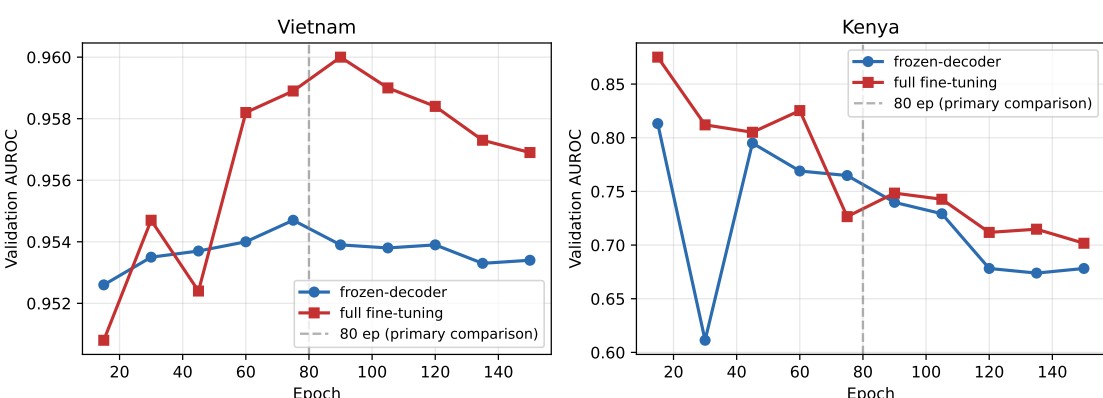

Figure 3: **Prithvi validation-AUROC convergence, 1–150 epochs (single seed, diagnostic).** Per-epoch test-set AUROC for Prithvi frozen-decoder and full fine-tuning on Vietnam (dense-positive, 18% positives) and Kenya (boundary case, 0.1% positives). Vertical dashed line at epoch 80 marks the primary-comparison budget. *Vietnam:* the frozen-decoder curve is flat from epoch 15 through 150 (drift within 0.002 AUROC); the full-FT curve peaks near epoch 90 (≈0.960) and slightly declines to 0.957 at epoch 150. *Kenya:* the full-FT curve peaks at epoch 15 (0.875) and declines to 0.702 by epoch 150, with small transient rebounds around epochs 60 and 90; the frozen-decoder is unstable and never recovers a stable value in this regime, ending at 0.678. In both regions and for both configurations, extending training past epoch 80 does not improve validation AUROC; for Kenya, longer training strictly hurts. Per-epoch evaluation is a post-hoc diagnostic; model selection remains last-epoch with no early stopping.

Table 15: **Spatial-pooling control (true-label AUROC).** A $5 \times 5$ windowed random forest (300-d) gains only about 0.01 over the per-pixel RF on average (Kenya drops by the same amount) and stays far below the best foundation model. This supports the interpretation that the GeoFM and U-Net advantage uses learned spatial context rather than naive local pooling. A per-pixel U-Net ablation ($1 \times 1$ convolutions, no spatial context) likewise falls to ≈RF (India 0.623, Vietnam 0.678).

| Region | per-pixel RF | windowed RF ($5 \times 5$) | best FM |
|---|---|---|---|
| India | 0.574 | 0.582 | 0.983 |
| Vietnam | 0.643 | 0.657 | 0.929 |
| Kenya | 0.651 | 0.641 | 0.853 |
| France | 0.695 | 0.703 | 0.961 |
| Netherlands | 0.815 | 0.827 | 0.961 |

Table 16: **FTW partial-labeling sensitivity (true-label AUROC).** Restricting negatives to high-confidence non-cropland (dropping ambiguous, possibly-unlabeled field pixels) leaves the frozen-GeoFM nearly unchanged in the three tested regions, while the spectral RF rises. This control does not indicate that FTW partial annotation drives the frozen-GeoFM result. France's RF gain ($0.695 \rightarrow 0.947$) reflects spectrally crop-like negatives (pasture, vineyards) that confuse a per-pixel RF; removing them leaves an easy task. The GeoFM was near-ceiling, so it barely moves.

| Region | RF (all neg) | RF (high-conf) | Prithvi (all neg) | Prithvi (high-conf) |
|---|---|---|---|---|
| India | 0.574 | 0.665 | 0.983 | 0.987 |
| France | 0.695 | 0.947 | 0.961 | 0.968 |
| Kenya | 0.651 | 0.698 | 0.766 | 0.774 |

Table 17: **Tile-disjoint sensitivity for the Table 2 frozen-probe column (true-label AUROC).** Each pair: chip-grouped (used in the main text) vs. MGRS-tile-disjoint AUROC for the same frozen probe. Prithvi-EO-2.0 is stable across both splits (drops $\leq 0.007$). TerraMind is also stable except in Kenya, where it drops from 0.745 to 0.698. AnySat drops on Kenya ($0.853 \rightarrow 0.580$), Vietnam ($0.897 \rightarrow 0.645$), and France ($0.943 \rightarrow 0.605$): its chip-grouped advantage may not reflect transferable feature quality. In Kenya, Prithvi (0.761) is more robust under tile-disjoint than AnySat (0.580). Cambodia is omitted (variant not run). Chip-grouped numbers preserve the per-chip statistical assumptions of the controlled comparison; this is the sensitivity check.

| | Prithvi-EO-2.0 | | TerraMind | | AnySat | |
| --- | --- | --- | --- | --- | --- | --- |
| Region | chip | tile | chip | tile | chip | tile |
| India | 0.983 | 0.984 | 0.967 | 0.969 | 0.939 | 0.945 |
| Vietnam | 0.929 | 0.922 | 0.916 | 0.915 | 0.897 | 0.645 |
| Kenya | 0.766 | 0.761 | 0.745 | 0.698 | 0.853 | 0.580 |
| France | 0.961 | 0.959 | 0.951 | 0.947 | 0.943 | 0.605 |
| Netherlands | 0.955 | 0.955 | 0.943 | 0.944 | 0.961 | 0.893 |

Table 18: **Threshold-based metrics complement AUROC** (mean over 10 seeds; F1/IoU at $p$=0.5; AP threshold-free). Dense-positive Cambodia and Vietnam show the three configurations close on IoU. France and Netherlands show the U-Net ahead on IoU, so the threshold metrics favor the supervised baseline more than AUROC alone would suggest. Kenya ($\sim 0.1\%$ positives): frozen decoder predicts all negatives (F1=IoU=0), full fine-tune is near all-negative, and U-Net retains AP=0.593, F1=0.159. "Frozen" = Prithvi backbone frozen, decoder trained.

| Region | Model | AUROC | AP | F1 | IoU |
| --- | --- | --- | --- | --- | --- |
| India | U-Net | 0.941 | 0.856 | 0.619 | 0.455 |
| India | Frozen | 0.982 | 0.970 | 0.557 | 0.386 |
| India | Full-FT | 0.985 | 0.974 | 0.604 | 0.433 |
| Cambodia | U-Net | 0.948 | 0.923 | 0.904 | 0.826 |
| Cambodia | Frozen | 0.947 | 0.912 | 0.919 | 0.850 |
| Cambodia | Full-FT | 0.949 | 0.914 | 0.923 | 0.856 |
| Vietnam | U-Net | 0.964 | 0.948 | 0.921 | 0.854 |
| Vietnam | Frozen | 0.955 | 0.927 | 0.912 | 0.839 |
| Vietnam | Full-FT | 0.958 | 0.933 | 0.917 | 0.847 |
| Kenya | U-Net | 0.891 | 0.593 | 0.159 | 0.086 |
| Kenya | Frozen | 0.653 | 0.304 | 0.000 | 0.000 |
| Kenya | Full-FT | 0.771 | 0.403 | 0.010 | 0.005 |
| France | U-Net | 0.977 | 0.963 | 0.942 | 0.891 |
| France | Frozen | 0.982 | 0.972 | 0.919 | 0.850 |
| France | Full-FT | 0.979 | 0.972 | 0.923 | 0.856 |
| Netherlands | U-Net | 0.981 | 0.977 | 0.895 | 0.809 |
| Netherlands | Frozen | 0.971 | 0.967 | 0.823 | 0.699 |
| Netherlands | Full-FT | 0.958 | 0.962 | 0.829 | 0.708 |

