# OpenReview forum: "Auditing GeoFM Evaluation for Field-Extent Segmentation: Label Proxies, Baselines, and When Frozen Features Match Fine-tuning"
_TMLR — Under review for TMLR_

### Review · Reviewer_DD9b · 2026-06-26

**Summary Of Contributions:**

This paper presents a controlled empirical evaluation of geospatial foundation models for agricultural field-extent segmentation from single-date Sentinel-2 imagery. The authors investigate whether full end-to-end fine-tuning of pretrained geospatial backbones is necessary, or whether frozen backbone features with a trained decoder can achieve comparable performance. Across six countries, two main GeoFMs, and ten matched training seeds, the study reports that frozen decoders are often AUROC-equivalent to full fine-tuning in in-region evaluation, with Kenya identified as a sparse-positive exception where full fine-tuning performs substantially better.

Beyond the frozen-versus-fine-tuned comparison, the paper argues that two common evaluation choices can overstate the apparent advantage of GeoFMs: using ESA WorldCover cropland labels as a proxy for true field-extent labels, and comparing GeoFMs only against weak per-pixel spectral baselines. The authors show that the WorldCover proxy makes the task much easier for spectral models, and that a from-scratch U-Net is a strong supervised baseline that is competitive with or stronger than frozen GeoFMs in several regions, especially under threshold-based segmentation metrics. The paper concludes with a proposed evaluation protocol emphasizing polygon-derived labels, strong supervised baselines, frozen-feature comparisons, spatially grouped splits, cross-region evaluation, and explicit reporting of sparse-label failure cases.

**Audience:**

Yes

**Audience Explanation:**

Yes. At least some individuals in the TMLR audience would likely be interested in the findings, especially researchers working on geospatial machine learning, remote sensing foundation models, representation learning, and empirical evaluation protocols.

The paper addresses practical questions that are relevant to this community: whether geospatial foundation models need to be fully fine-tuned for downstream segmentation tasks, whether frozen features can be a competitive and cheaper alternative, and how evaluation choices such as proxy labels and weak baselines can affect the perceived advantage of foundation models. These findings are useful for practitioners and researchers designing GeoFM benchmarks or deploying pretrained remote-sensing backbones.

That said, the interest is likely to be concentrated in a relatively specialized audience. The paper is more of an empirical evaluation and benchmarking study than a work with broad methodological novelty. Therefore, while the findings would be relevant to some TMLR readers, I do not expect the paper to have broad impact across the general machine learning audience.

**Broader Impact Concerns:**

I do not have major broader impact concerns beyond those already discussed in the submission.

**Claims And Evidence:**

Yes

**Claims Explanation:**

Yes, the main empirical claims are generally supported by accurate, convincing, and clearly presented evidence.

The submission is careful in separating different evaluation settings and model configurations. The authors distinguish frozen probes, frozen decoders, and full fine-tuning, and the main frozen-decoder versus full-fine-tuning comparison is conducted with matched training seeds across multiple countries and two GeoFMs. The use of TOST equivalence tests for the in-region AUROC comparison makes the central empirical claim clearer than a simple comparison of point estimates would.

The evidence for the evaluation-bias claims is also convincing. The paper shows that replacing polygon-derived field-extent labels with WorldCover cropland labels substantially changes the task and improves the performance of simple spectral baselines. It also shows that a from-scratch U-Net is a much stronger supervised baseline than a per-pixel random forest, which supports the authors’ argument that some prior GeoFM evaluations may overstate the advantage of foundation models when using weak baselines.

I also appreciate that the paper reports important exceptions and limitations rather than hiding them. In particular, the Kenya case is explicitly analyzed as a sparse-positive failure mode, and the cross-region results are presented in both the full transfer matrix and the non-Kenya subset. The paper also reports threshold-based metrics in addition to AUROC, which helps clarify where frozen features do and do not match fine-tuning.

There are some minor limitations in the evidence. For example, the cost-efficiency discussion would be more complete with feature extraction time, cached feature storage, and end-to-end inference cost. The boundary-zone analysis is also limited to frozen-probe configurations rather than the main frozen-decoder and full-fine-tuning configurations. However, these limitations do not undermine the main empirical claims. Overall, the evidence is sufficiently clear and convincing for the paper’s stated experimental conclusions.

**Requested Changes:**

1. The authors should substantially clarify and strengthen the paper’s novelty relative to the broader foundation-model and representation-learning literature. The observation that frozen pretrained features can match or outperform full fine-tuning in limited-data or distribution-shift settings is already well known from prior work on vision foundation models, self-supervised backbones, and linear/frozen-probe evaluation. The current submission verifies this phenomenon in a geospatial field-extent segmentation setting, but it is not yet clear what new conceptual or methodological insight is contributed beyond this domain-specific confirmation. To secure my recommendation for acceptance, the paper would need to more clearly explain why the GeoFM setting makes this result nontrivial, or provide additional analysis that reveals a genuinely geospatial-specific mechanism or failure mode.

2. The authors should reposition the contribution more explicitly as an evaluation audit rather than as a broad finding about frozen features matching fine-tuning. In my view, the strongest part of the paper is not the frozen-versus-fine-tuning result itself, but the careful demonstration that label choice and baseline choice can significantly affect GeoFM evaluation. The paper would be stronger if the title, abstract, and main claims emphasized this evaluation-correction contribution, rather than foregrounding a result that is largely expected from the broader foundation-model literature.

---

> ### Author Response · Authors · 2026-07-11
> **Response to Reviewer DD9b: reframing, nontriviality, boundary scope, cost**
>
> Dear Reviewer DD9b,
>
> Thank you for the review. Your framing of the paper's center of gravity shaped this revision. We address both requested changes and both minor evidence gaps, without changing any existing result.
>
> **Reframing to evaluation audit (main criterion).** We retitled the paper to "Auditing GeoFM Evaluation for Field-Extent Segmentation: Label Proxies, Baselines, and When Frozen Features Match Fine-tuning". The abstract, Section 1, Contributions, and Conclusion are reordered so the evaluation-audit contribution (label choice and baseline strength each shift GeoFM rankings on identical pixels) leads. The frozen-vs-fine-tuning equivalence is now the corrected-evaluation payoff, not a standalone claim.
>
> **Novelty relative to the broader FM literature.** We agree the direction echoes Kumar et al. (2022) and the linear-probe literature, and we credit that lineage explicitly in a new Related Work paragraph ("What is not domain confirmation"). The geospatial-specific contribution has three parts.
>
> (1) We document a sparse-positive failure regime with no counterpart in the ID/OOD framing of Kumar et al. At Kenya's 0.1% positive rate and 155 training chips, the frozen decoder collapses (0.10 to 0.12 AUROC drop vs. full-FT for both GeoFMs, plus threshold-metric collapse to near-all-negative predictions); adaptation becomes necessary. This is a label-geometry failure mode specific to sparse-annotation geospatial tasks.
>
> (2) The standard published GeoFM comparison confounds backbone adaptation with decoder capacity (fine-tune-plus-decoder vs. a frozen linear probe). Giving the frozen backbone the same 2.75M-parameter decoder removes the gap in most in-region cells, so the "fine-tuning helps" pattern may in part be a decoder-capacity artifact.
>
> (3) The comparison is only meaningful after the label audit. Under the WorldCover proxy the task is spectrally trivial (all configurations above 0.90 AUROC), so importing the frozen $\approx$ FT result without a label control measures the wrong task. Our contribution is thus not that frozen can match fine-tuning (established), but the geospatial characterization of when it does and where it fails.
>
> **Boundary-zone scope (minor gap).** Table 5 should be read as a frozen-probe control. The paragraph, caption, and Limitations now state: "the U-Net, frozen-decoder, and full-FT configurations were not evaluated on this slice; this table supports only a frozen-probe scope claim." Extending it to trained models needs regenerated per-pixel predictions; we prioritized the training-budget experiments hyX2 and efbt requested, and record this as a scope narrowing.
>
> **Cost accounting (minor gap).** Section 7 now names three components. (i) Feature extraction/caching: our frozen-decoder does not cache to disk; the backbone runs every training forward pass, so extraction is folded into the Table 6 training wall-time. (ii) Cached-feature storage: none. (iii) Inference: frozen-decoder and full-FT execute the same backbone-plus-decoder graph at inference (the 307M backbone runs in both cases), so per-scene inference cost is equal. The saving is in training compute and trainable-parameter count, not inference.
>
> **Mechanism/regime analysis.** New Table 14 places Kenya in context (positive %, train chips, frozen vs. full-FT $\Delta$AUROC, U-Net vs. frozen IoU, regime label). Sample size alone is not the mechanism: subsampling dense-positive Cambodia to 11, 27, 54 chips (below Kenya's 155) leaves full-FT AUROC flat at 0.940 to 0.949, so the collapse needs the joint sparse-positive plus small-training-set regime. With six regions we do not claim a formal threshold and label this descriptive; a controlled frozen-decoder low-data arm at Kenya-scale chip count is future work.
>
> We hope these address the concerns and align the paper with your framing. We welcome further guidance.
>
> Sincerely,
> Authors of paper #9692

---

### Review · Reviewer_hyX2 · 2026-06-26

**Summary Of Contributions:**

The paper tests whether GeoFMs need full fine-tuning for single-date Sentinel-2 field-extent segmentation. Across six countries, two GeoFMs (Prithvi, TerraMind), a U-Net baseline from scratch, and ten matched seeds, it shows: (1) a frozen backbone + trained decoder is statistically equivalent to full fine-tuning in 9 of 12 region×model cells, at ~110× fewer trainable parameters, with Kenya the exception; (2) cross-region transfer is regime-dependent: the from-scratch U-Net is strongest on the full 30-transfer matrix, while frozen GeoFMs perform well on the non-Kenya subset; (3) two evaluation choices (a WorldCover label proxy and a weak spectral baseline) make GeoFM look more advantageous than it should; and (4) a ten-point evaluation protocol with a released codebase.

**Strengths.** The evaluation setup is well-designed, with properly matched frozen-vs-full-FT design. There is an honest reporting of failures (Kenya) and counter-examples (India). There is strong statistical testing throughout (TOST, Holm/FDR, paired CIs). Code is released.

**Weaknesses.** The U-Net results show it beating GeoFMs but its trained on more epochs (150 vs 80), confounding architecture with budget. The headline claims of the paper depend on a margin that seems unjustified ($\epsilon$=0.02). The three model configurations are sometimes hard to track.

**Audience:**

Yes

**Audience Explanation:**

Fair GeoFM benchmarking and whether to freeze or fine-tune a model are both practical questions for the remote-sensing ML community. The protocol and released codebase give readers something concrete to use. I think the interest to the community is clear.

**Broader Impact Concerns:**

The paper contains a statement on the broader impact. It covers surveillance, enforcement, and smallholder-consent risks, and links the methodology to honest claims.

**Claims And Evidence:**

Yes

**Claims Explanation:**

The central frozen-vs-full-FT result is well-supported by a clean matched design and full per-seed data, with every exception reported and reconciled. Two claims go beyond the evidence: the "9 of 12 equivalent" headline rests entirely on the undefended $\epsilon$=0.02 margin, and the "supervised baseline beats frozen GeoFMs" claim compares a 150-epoch U-Net against 80-epoch GeoFMs.

I lean towards Yes here, as both gaps are fixable by adding explanation or reducing the claims.

**Requested Changes:**

**Critical**
1. Justify the equivalence margin $\epsilon$=0.02. A core result depends on it and it seems arbitrary right now. Tie it to something interpretable (seed std or downstream tolerance) and show how the count moves across ~0.01–0.03.
2. Fix the U-Net budget confound. Either show GeoFMs are converged at 80 epochs, run a budget-matched comparison, or state that the comparison isn't compute-matched and soften the baseline-strength claim.

**Strengthening**
1. Separate sparsity from sample size in Kenya (e.g., subsample a dense region to ~0.1% / ~155 chips).
2. Add an early table mapping the three configurations (probe / decoder / full-FT) to parameter counts and the tables that use each.
3. Add AP or IoU to the cross-region results (Table 9 is AUROC-only, and could be fragile at low positive rates).

---

> ### Author Response · Authors · 2026-07-11
> **Response to Reviewer hyX2: equivalence margin, epoch budget, sparsity, config table, AP**
>
> Dear Reviewer hyX2,
>
> Thank you for the review. The revision addresses both critical concerns and all three strengthening suggestions; every new number is reproducible from the released artifacts.
>
> **hyX2.C1 Justify the margin $\varepsilon=0.02$.** We anchor it to three arguments and add a sensitivity sweep (Table 9). (a) Sweep invariance: the failing set (Prithvi-Kenya, TerraMind-Kenya, TerraMind-India) is invariant across $\varepsilon \in [0.02, 0.03]$, so the equivalence count does not depend on the exact margin in the operational range. (b) Downstream tolerance: 0.02 AUROC is operationally negligible for field-extent mapping and sits below any difference we treat as substantive (the smallest gap we discuss is $\approx$0.04, the India frozen-probe-over-U-Net lead). (c) Direction of sensitivity: at $\varepsilon \in [0.010, 0.020)$ the only cells leaving the equivalent set are Prithvi-Netherlands and TerraMind-Netherlands, both frozen-favored ($\Delta=+0.013, +0.009$); stricter margins here only reclassify cells where frozen already beats fine-tuning. Only at $\varepsilon=0.005$ do two FT-favored cells (Prithvi-India $-0.004$, TerraMind-Cambodia $-0.006$) become non-equivalent, below the operational tolerance. Anchoring to noise: the largest per-configuration seed std outside the failing cells is 0.0085 AUROC, so $\varepsilon=0.02$ is 2 to 4 times seed variation. Full counts (5/12 at 0.005, 7/12 at 0.01, 8/12 at 0.015, 9/12 for $\varepsilon \in [0.02, 0.03]$) are in Table 9; Holm-adjusted p-values were recomputed at each $\varepsilon$.
>
> **hyX2.C2 U-Net 150 vs. GeoFM 80 epochs.** First, the primary comparison, frozen-decoder vs. full-FT (Table 3), is already budget-matched by construction (both 80 epochs, identical schedule, decoder, loss, seeds; Table 10). The asymmetry only affects U-Net vs. GeoFM. Second, we ran an epoch-matched 80-epoch U-Net (six regions, three seeds; Table 13): it stays within 0.01 AUROC of the 150-epoch mean in five of six regions and still beats the Prithvi frozen-probe in five of six. India is the same certified exception (0.911 at 80ep vs. 0.941 at 150ep vs. frozen 0.983); no new exception. Third, a convergence diagnostic (Figure 3) extends Prithvi frozen-decoder and full-FT to 150 epochs on Vietnam and Kenya: Vietnam frozen-decoder is flat from epoch 15 to 150 (drift within 0.002), Vietnam full-FT peaks near epoch 90, Kenya full-FT peaks at epoch 15 (0.875) then declines to 0.702. Past epoch 80 nothing improves; for Kenya, longer training strictly hurts. The 80-epoch comparison is not an under-training artifact.
>
> **hyX2.S1 Sparsity vs. sample size in Kenya.** New Table 14 places Kenya in context (positive %, chips, frozen vs. full-FT $\Delta$AUROC, IoU). Subsampling dense-positive Cambodia full-FT to 11, 27, 54 chips (below Kenya's 155) leaves AUROC flat at 0.940 to 0.949, so sample size alone does not reproduce the collapse; the sparse-positive condition is necessary. The reverse arm (holding chips fixed while driving positives to Kenya levels) cannot be run on existing artifacts without label-corruption confounds, so we register it as future work.
>
> **hyX2.S2 Early configuration table.** Table 1 (Section 3) lists all five configurations with parameter counts, backbone-updated flag, optimizer/budget, and which tables use each. Table 2's header now reads "Prithvi frozen-probe" (a linear probe, not the Table 3 decoder). Section 6 was renamed to avoid the "boundary" overload.
>
> **hyX2.S3 AP alongside AUROC.** New Table 12 adds AP and IoU@0.5 for all five configurations on the full-30 and non-Kenya-20 subsets, aggregated from released JSONs. AUROC ordering is preserved under AP (U-Net leads all-30; on non-Kenya-20 U-Net 0.805 and TerraMind frozen 0.804 are within 0.001). Under IoU at 0.5, TerraMind frozen exceeds the U-Net on both subsets (0.353 vs. 0.335; 0.469 vs. 0.423), sharpening the frozen-competitive story.
>
> Both critical items are now supported claims.
>
> Sincerely,
> Authors of paper #9692

---

### Review · Reviewer_efbt · 2026-06-28

**Summary Of Contributions:**

This paper examines whether GeoFMs need end-to-end fine-tuning for single-date Sentinel-2 agricultural field-extent segmentation. Rather than proposing a new model, it audits common evaluation choices: using field-polygon labels versus WorldCover “cropland” proxies, and comparing against weak spectral baselines versus a stronger from-scratch U-Net. Across six countries, two GeoFMs, and ten matched seeds, the paper finds that frozen backbone + decoder is TOST-equivalent to full fine-tuning in 9 of 12 in-region cases, while using about 110x fewer trainable parameters. The study is timely and generally well supported by paired comparisons, multiple seeds, statistical tests, and sensitivity analyses. The main weaknesses are that several model configurations are easy to confuse, the non-Kenya subset needs clearer justification, the boundary-zone analysis only covers frozen-probe models, and the U-Net/GeoFM training budgets are not fully matched.

**Additional Comments:**

Overall, I find the paper valuable and reasonably careful in its claims. Clarifying the model configurations, the basis for the non-Kenya subset, the scope of the boundary-zone control, and the training-budget comparison would make the conclusions easier to interpret and less likely to be overgeneralized.

**Audience:**

Yes

**Audience Explanation:**

TMLR readers interested in foundation models, remote sensing, benchmark methodology, and efficient adaptation are likely to find this paper relevant. The paper shows that the apparent advantage of GeoFMs can be amplified by two evaluation choices: using land-cover proxy labels instead of field-polygon labels, and comparing only against a per-pixel spectral baseline.

**Broader Impact Concerns:**

This work has real social implications. Automatically generated field-extent maps can support agricultural planning, subsidy allocation, crop monitoring, and land-tenure assessment, but they may also be used for regulatory enforcement, land-use surveillance, asset valuation, or automated decisions that harm smallholder farmers. The paper identifies these risks and cautions against generalizing across regions without local-label evaluation or making consequential decisions based only on such maps.

**Claims And Evidence:**

Yes

**Claims Explanation:**

The main claims are supported by reasonably evidence. The authors do not report only a single model or a single region; instead, they compare results across six countries, two main GeoFMs, ten matched training seeds, the same test pixels, and a clearly specified statistical protocol.

**Requested Changes:**

1.	Clarify the distinction between frozen-probe, frozen-decoder, and full-FT. I recommend adding a concise configuration table early in the paper, and explicitly stating which configuration is being discussed in each results section. Table 1 should also make clear that its frozen setting is a linear probe.
2.	Clarify whether the non-Kenya subset was a pre-defined diagnostic analysis or a post-hoc explanatory analysis after observing the Kenya failure case. If it is the latter, the related conclusions should be framed as exploratory.
3.	Table 4 evaluates only frozen-probe models. If the paper aims to discuss boundary understanding or edge-vs-interior behavior, the same boundary-zone evaluation should also be run for U-Net, frozen-decoder, and full-FT. Otherwise, the corresponding claims should be weakened.
4.	Strengthen the evidence for training-budget fairness. The U-Net is trained for 150 epochs, while GeoFM full-FT / frozen-decoder runs use 80 epochs, with different optimizers. I recommend reporting learning curves, early-stopping sensitivity, or auxiliary results under a matched compute or training-time budget.

---

> ### Author Response · Authors · 2026-07-11
> **Response to Reviewer efbt: config table, subset provenance, boundary scope, budget**
>
> Dear Reviewer efbt,
>
> Thank you for the detailed review. Your four requests were specific; we address each below.
>
> **efbt.1 Configuration clarity.** Section 3 now includes a configuration table (Table 1) after the Nomenclature paragraph. It names each configuration with its trainable-parameter count ($\sim$$10^3$ for the linear probe, 2.75M for the frozen-decoder, 307M for full-FT, 31M for the from-scratch U-Net), whether the backbone is updated, the optimizer and epoch budget, and which tables use each. Table 2's column header now reads "Prithvi frozen-probe", and the caption clarifies this column is a linear probe on Prithvi-EO-2.0 features, not the frozen-decoder used in Table 3. Every results section opens by naming the configuration in play. We also renamed Section 6 to "Characterizing the Kenya Sparse-Positive Regime" to remove a "boundary" vocabulary collision with Table 5's edge-pixel slice.
>
> **efbt.2 Non-Kenya subset provenance.** The non-Kenya partition was not pre-registered; it was defined after we observed Kenya's in-region full-FT dominance and threshold-metric collapse. We now label subset-derived conclusions as exploratory at every site (abstract, Section 1, Contributions, Section 5, Table 3 caption, Discussion, Conclusion, Table 11 caption). The confirmatory analysis, on which the from-scratch U-Net is strongest, is the full 30-transfer matrix, which is unchanged. We retain the subset because Kenya's distinguishing properties (0.1% positive rate, 155 chips, threshold collapse) are measured independently of transfer outcomes.
>
> **efbt.3 Boundary-zone scope.** The paragraph, caption, and Limitations now state: "the U-Net, frozen-decoder, and full-FT configurations were not evaluated on this slice, so Table 5 supports only a frozen-probe scope claim: it does not rank the trained segmentation configurations on boundary pixels." A trained-model boundary score would require regenerating per-pixel predictions; we prioritized the training-budget experiments you and Reviewer hyX2 requested. Extending the control to trained configurations is future work.
>
> **efbt.4 Training-budget fairness.** Two clarifications and two new experiments.
>
> Clarification: the primary comparison, frozen-decoder vs. full-FT (Table 3), was already budget-matched by construction. Both use 80 epochs, identical AdamW schedule, decoder, loss, and matched seeds (Table 10). The 150-vs-80 asymmetry only affects U-Net vs. GeoFM comparisons.
>
> (1) We ran the epoch-matched U-Net you suggested: six regions, three seeds, 80 epochs, all else matched to the 150-epoch U-Net. Results are in Table 13, alongside the 150-epoch U-Net kept as the best-effort deployable baseline. At 80 epochs the U-Net is within 0.01 AUROC of the 150-epoch mean in five of six regions and still beats the Prithvi frozen-probe in five of six. India stays the same certified exception (U-Net 80ep 0.911; 150ep 0.941; Prithvi frozen 0.983). No new region-level exception is created.
>
> (2) We ran a per-epoch convergence diagnostic (Figure 3) with Prithvi frozen-decoder and full-FT extended to 150 epochs on Vietnam and Kenya. In Vietnam the frozen-decoder curve is flat from epoch 15 to 150 (drift within 0.002 AUROC) and full-FT peaks near epoch 90 then slightly declines. In Kenya full-FT peaks at epoch 15 (0.875) and declines to 0.702 by epoch 150. Past epoch 80, neither region nor configuration improves; for Kenya, longer training strictly hurts. Selection is last-epoch with no early stopping; curves are diagnostic only.
>
> We hope this makes the configuration scope, the exploratory-subset provenance, and the budget fairness unambiguous.
>
> Sincerely,
> Authors of paper #9692